https://doi.org/10.1038/s41467-021-25824-9　　**OPEN**

# Community and single cell analyses reveal complex predatory interactions between bacteria in high diversity systems

Yossi Cohen[1], Zohar Pasternak[1,2], Susann Müller [3], Thomas Hübschmann[3], Florian Schattenberg[3], Kunjukrishnan Kamalakshi Sivakala [1], Alfred Abed-Rabbo[4], Antonis Chatzinotas [3,5,6] & Edouard Jurkevitch [1✉]

A fundamental question in community ecology is the role of predator–prey interactions in food-web stability and species coexistence. Although microbial microcosms offer powerful systems to investigate it, interrogating the environment is much more arduous. Here, we show in a 1-year survey that the obligate predators *Bdellovibrio* and like organisms (BALOs) can regulate prey populations, possibly in a density-dependent manner, in the naturally complex, species-rich environments of wastewater treatment plants. Abundant as well as rarer prey populations are affected, leading to an oscillating predatory landscape shifting at various temporal scales in which the total population remains stable. Shifts, along with differential prey range, explain co-existence of the numerous predators through niche partitioning. We validate these sequence-based findings using single-cell sorting combined with fluorescent hybridization and community sequencing. Our approach should be applicable for deciphering community interactions in other systems.

[1] Department of Plant Pathology and Microbiology, Institute of Environmental Sciences, Faculty of Agriculture, Food and Environment, The Hebrew University of Jerusalem, Rehovot 76100, Israel. [2] Division of Identification and Forensic Science, Israel Police, National Headquarters, Jerusalem, Israel. [3] Department of Environmental Microbiology, Helmholtz Centre for Environmental Research - UFZ, Permoserstrasse 15, 04318 Leipzig, Germany. [4] Faculty of Science, Bethlehem University, Bethlehem, Palestine. [5] Institute of Biology, Leipzig University, Talstrasse 33, 04103 Leipzig, Germany. [6] Centre for Integrative Biodiversity Research (iDiv) Halle-Jena-Leipzig, Deutscher Platz 5e, 04103 Leipzig, Germany. ✉email: edouard.jurkevitch@mail.huji.ac.il

A fundamental question in community ecology is the role of predator–prey interactions in trophic web stability and species coexistence. This is central to the understanding of how food webs, which are the basis of sustainable life, are maintained. Predatory interactions have profound effects on food webs, among others by promoting diversity and creating trophic cascades that may vary in length and strength, depending on various parameters, e.g. predator size, temperature and else[1,2]. Inherent properties such as the prey range of a predator, i.e. being a generalist or a specialist, or spatial complexity further strongly affect the type of control, promoting top-down or bottom-up effects[3–5].

In aquatic food webs, protists (which are mostly considered generalists) are essential consumers of bacteria, and along with bacteriophages (mostly specialists) they may be the largest contributors to bacterial mortality and turnover[6,7]. However, the ecological role of other members of the predatory guild has been much less explored[5]. The *Bdellovibrio* and like organisms (BALOs) are highly motile gram-negative bacteria that obligatorily prey upon other gram-negative cells[8]. They are smaller than their prey, which they mainly consume after penetrating their periplasmic space, to grow as a multi-nucleoid cell which will divide into the flagellated progeny, and exit the remains of the prey[8]. BALOs are ubiquitous in soil and in water bodies and are considered to be "intermediate" versatilists, neither generalists nor specialists[9]. Indeed, isolated BALO strains tested with prey arrays show utilisation ranging from single[10] to many prey strains[11,12]. In natural microcosms, specific BALO predators are controlled by the availability of an adequate prey[9] and rapidly respond to prey abundance[13]; in turn, spiking microcosms with a specific BALO revealed that the predator impacted upon the bacterial community structure[14]. Thus, BALOs may constitute a "sideway control" (i.e. neither top-down nor bottom-up) over food webs, affecting bacterial community structure and contributing to community succession[15].

However, such microcosm experiments, which were also limited in time, cannot provide the accurate identity of interacting predators and prey in nature, i.e. the prey ranges of individual predators in the BALO community. This information is essential if one is to understand their impact on microbial communities, predator–prey dynamics, what sustains their diversity, and to potentially manage in situ interactions for ecological and biotechnological applications. High throughput sequencing technologies may uncover the diversity of BALOs and along with quantitative PCR, network computing, and in situ detection approaches enable their identification and quantification without relying on bottlenecks created by isolating BALOs on particular prey[16,17].

Here, we specifically analysed the community dynamics of the two major BALO clades (the Bdellovibrionales and the Bacteriovoracales) and their association with prey over a year, at three wastewater treatment plants (WWTPs), and invoke predator–prey theory to explain our results. WWTPs are crucial for keeping public health and reducing environmental pollution[18] and being the most microbe-diverse human-made habitat, they sustain numerous interactions, including predatory interactions[14,19,20]. We thus asked whether this diversity is also found in BALOs, what the predators' natural prey are and what mechanisms explain their co-existence. We hypothesized that niche differentiation sustains predator diversity by prey range partitioning, temporal differentiation[21,22] and through fluctuating predator and prey populations. Finally, in order to validate sequence-based computing results, we developed a direct approach based on FISH tagging[23] and cell sorting to identify the interacting predators and prey.

## Results

**BALO diversity and phylogeny.** Diversity indices of the Bdellovibrionales (Bd) and of the Bacteriovoracales (Bx) show that Bd and Bx diversity did not differ between the floc and the liquor fractions at each site but they were always higher in the former (Table S1). The relative abundance of the predators in the total bacterial community, although fluctuating, was generally higher for Bd than for Bx (Figure S1), and while Bd were always present, Bx were absent from some samples. As sequence-based abundance analysis is only relative and may also be biased[24], a targeted 16S rRNA gene copy-based qPCR analysis of Bd and Bx absolute abundance was performed on the Shafdan (SH) samples. It showed no significant differences in absolute abundance over the time series between the two clades, averaging $0.5 \pm 0.2\%$ of the total bacterial population, for each (Figure S2). Furthermore, Bd and Bx OTU rank abundance (OTU abundance $> 0.1\%$) patterns in each fraction (floc, liquor) at each site (SH, Al-Bireh (AB), Langenreichenbach (LB)) were similar (pairwise Kolmogorov–Smirnov test, $p > 0.05$). OTU composition of the Bx community was more diverse between samples at each site and within each fraction than that of the Bd community, as reflected by larger Bray–Curtis dissimilarities (Figure S3A). This may explain the greater Bx richness over the time series, as compared to Bd (573 vs 133 OTUs, respectively). Taken together, the data suggest that while diversity differed between the two BALO clades, these appear to be similarly abundant.

A maximum likelihood analysis revealed a large, hitherto undetected phylogenetic diversity in both the Bd and the Bx (Fig. 1). Strikingly, most OTUs were in clusters separated from those containing previously characterised cultured strains. The Bd tree contained two deeply separated groups, while Bx OTUs formed two large, separated sister branches (Fig. 1A, B). Only a single of the most abundant Bd OTUs clustered near a cultured strain (*Bdellovibrio* strain W); in Bx, only a single OTU was found in the branch constituted of known strains; all the other Bx OTUs obtained in this study formed a second, very diverse cluster which included the *Bacteriovorax stolpii* type strain. Thus, the BALOs populating WWTPs while diverse are almost not represented in cultures. A correlation analysis (see below) linking predator and prey OTUs revealed that prey range was not linked to predator phylogeny, i.e. predators did not "specialize" in predating upon specific taxa, whether the predator had a wide ($> 20$ prey) or a narrow (1–5 prey) prey range (Supplementary Data 1).

**BALO spatial distribution between WWTPs.** A comparison between WWTPs showed that Bd and Bx community structure dissimilarity significantly increased with geographical distance ($p$-value $< 0.001$; chance-corrected within group agreement $A = 0.08–0.236$, Figure S3B). The differences were smaller between SH and AB than between each of these and the distant LB WWTP (Figure S3B). Accordingly, more OTUs were shared between SH and AB than between SH or AB and LB (Figure S4). Although a large proportion of the OTUs in both clades were shared between the three plants, Bd and Bx were differently distributed among the WWTPs: SH and AB Bd OTUs were almost all shared (113/123, 91.8%), and mostly shared between the three sites (90/133, 67.7%), with only a few OTUs being site-specific; As for Bx, less OTUs were common to SH and AB (151/403, 37.5%) or shared between the three sites (99/573, 17.3%) (Figure S4). Yet, and although their abundance varied between sites and in time within a site, the 20 most common OTUs of each predator in each fraction, which constituted 88 to 98% of the total Bd and Bx reads, respectively, were almost all ubiquitous, i.e. shared between the three WWTPs (Figure S5).

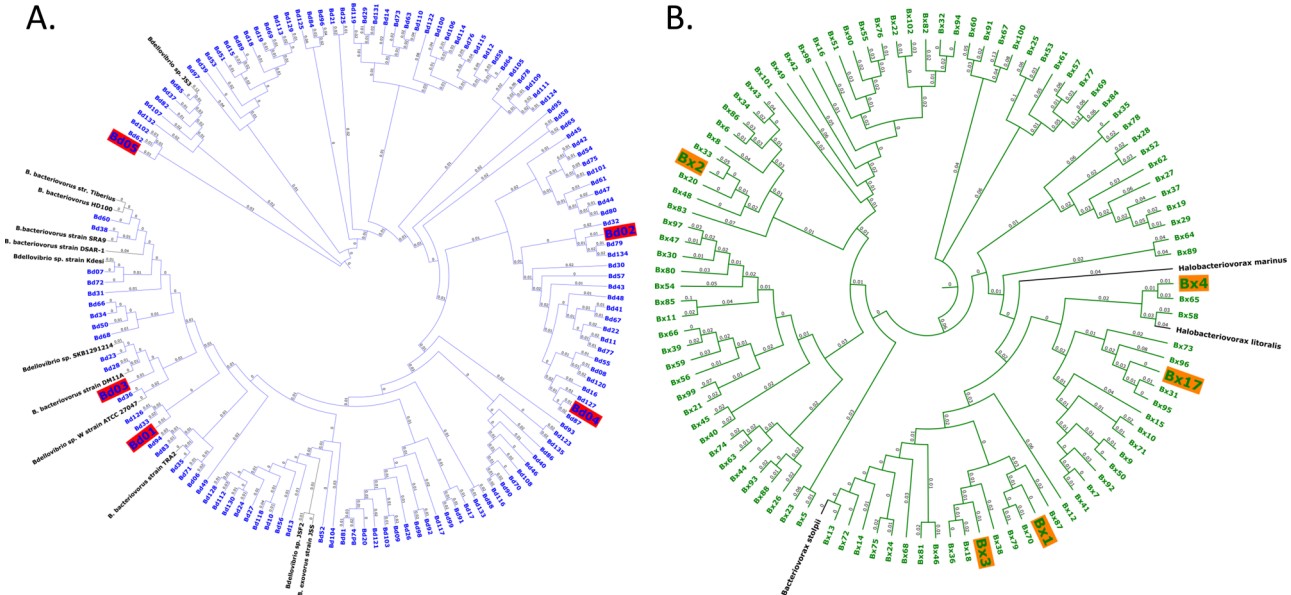

**Fig. 1 Phylogenetic analysis of BALOs.** Molecular phylogenetic analysis of the 100 most abundant 16S rRNA gene sequence-based OTUs of each of the two predatory bacterial families, Bdellovibrionales (**A**) (the Bd OTUs identified in this study are in blue fonts) and Bacteriovoracales (**B**) (the Bx OTUs identified in this study are in green fonts), using maximum likelihood. Distances (0–1) are written on the branches. The labelled OTUs are the five most abundant OTUs of the predators in the different WWTPs.

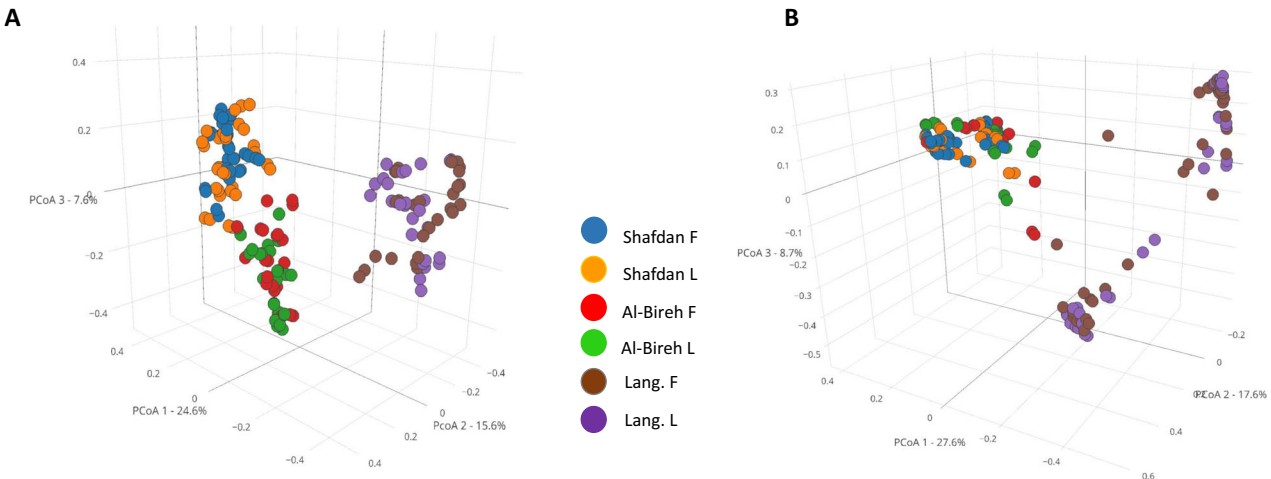

**Fig. 2 Floc and liquor BALO community structure.** PCoA of Bdellovibrionales (**A**) and Bacteriovoracales (**B**) community distribution of the floc (F) and liquor (L) fractions at the Shafdan (SH), Al-Bireh (AB), and Langenreichenbach (LB) WWTPs, using Bray–Curtis distances according to 16S rRNA gene community composition. Each symbol represents a single time point.

**Predator dynamics**. In total, flocs supported a higher population of predators than the liquor fraction, probably due to the larger bacterial populations associated with the particles (Figure S2) but flocs and liquor BALO populations did not segregate spatially, i.e. no associations of specific BALO populations (OTUs) with flocs or with the liquor were detected (Fig. 2). Nevertheless, BALOs in these two microhabitats exhibited different yearlong dynamics, reflected in the differential abundance of specific OTUs over time (e.g. Bd10, Bx5 and Bx8 at SH; Bx4 and Bx5 at AB; Bd18, and Bx20 at LB), oscillating between dominance and falling below detection levels (Fig. 3). Some of these oscillations had low frequencies and appeared to be linked to temperature upon seasonal changes (Table S2, $|r| = 0.61 \pm 0.12$; $0.7 \pm 0.27$ for Bd and Bx respectively); no other parameter consistently correlated with fluctuations in the predatory populations. Some predators were present during the cooler parts of the year (Bd7, Bd9, Bd13, Bx3, Bx5), others only during the warmest periods (Bx6, Bx7, Bx20).

Still, others were present all year long (Bd1, Bd2, Bd5, Bx1). Oscillations also occurred on a much shorter time frame, within weeks or less (e.g. Bx12, Bx11, Bx7 at SH, AB, and LB, respectively, H, I, L) (Fig. 3).

**Co-occurrence networks**. We thus postulated that short-term oscillations may be driven by local effects such as prey availability. In order to link predator population dynamics to those of potential prey, a negative Kendall correlation analysis was performed between potential prey OTUs and predator OTUs, under the assumption that sustained predation will decrease prey abundance, and vice versa[25,26]. OTUs from the dataset of a previous study using the same samples[22] were identified by taxonomic affiliation, and those affiliated with gram-positive clades (Actinobacteria, Firmicutes) were removed to restrict false-positive correlations resulting from unlinked or indirect interactions, leaving potential gram-negative prey. The analysis, based

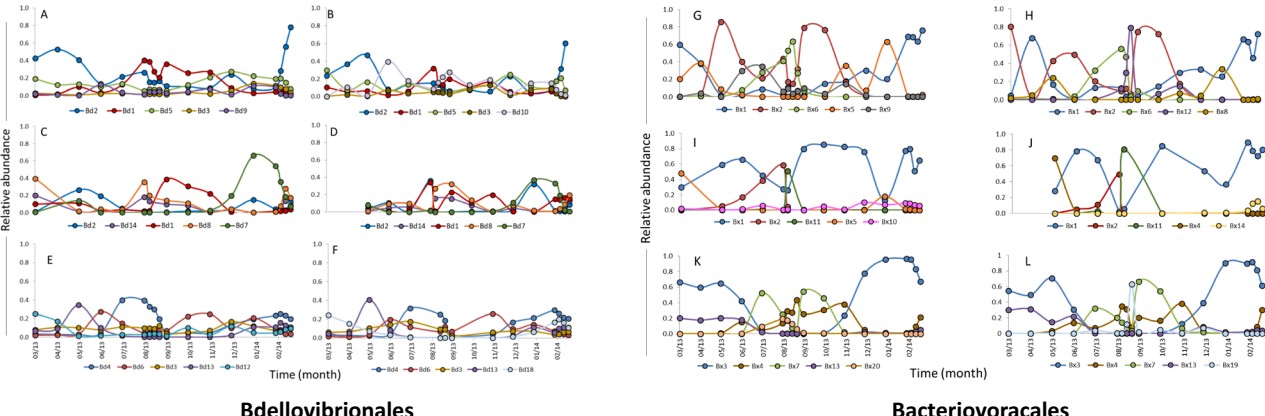

**Fig. 3 Yearly dynamics of BALOs.** Annual dynamics of the five most abundant Bdellovibrionales (**A**–**F**) and Bacteriovoracales (**G**–**L**) OTUs at the Shafdan (**A**, **B**, **G**, **H**), Al-Bireh (**C**, **D**, **I**, **J**), and Langenreichenbach (**E**, **F**, **K**, **L**) WWTPs over a year, with 18 sampling times. A, C, E, G, I, K: flocs. B, D, F, H, J, L: liquor. Relative abundances are based on read numbers.

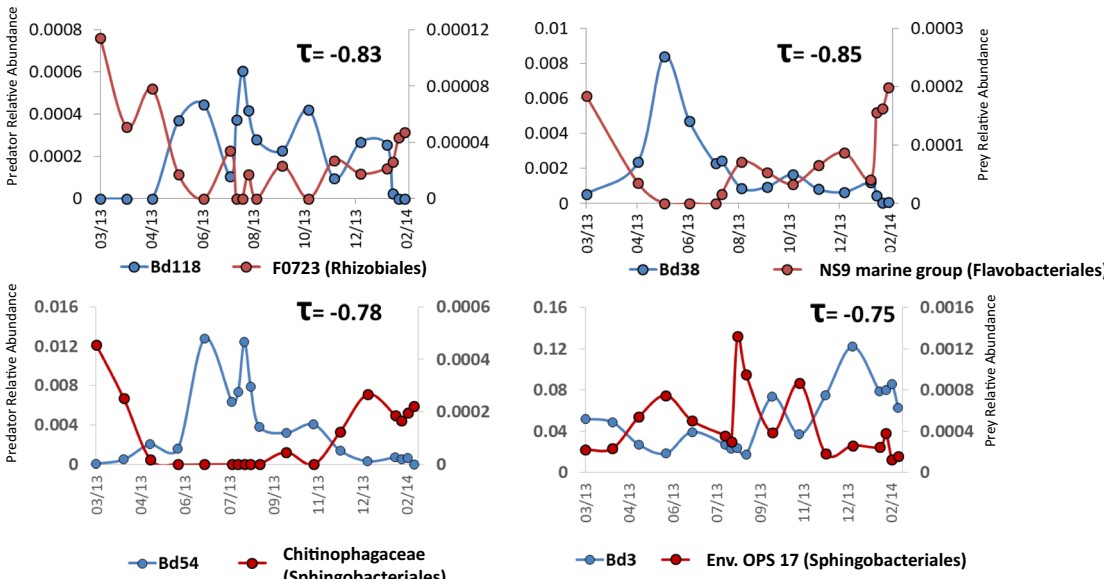

**Fig. 4 BALO-prey dynamics.** Negative correlations (Kendall $\tau$ > -0.7, $p$ < 0.05) of predators with potential prey identified at the family level (the order is in parentheses).

on significant negative correlations between the three annual datasets (total bacteria, Bdellovibrionales and Bacteriovoracales) ($< -0.7$, $p$-value $< 0.05$) sieved $>99.5\%$ of possible connections, revealing the co-occurrence of short time scale oscillations between predators and gram-negative populations, exhibiting patterns akin to predator–prey cycles, defining potential predator–prey interactions (Fig. 4). Networks were then constructed, generating strong potential predator–prey links with high confidence (Table S3). The high values of the modularity index of the networks (0.54 to 0.86) suggest a robust modular structure[27]. The number of connections per node largely varied (Shafdan: $4.52 \pm 0.76$, $2.41 \pm 0.32$ for Bd; $3.57 \pm 0.61$, $2.18 + 0.41$ for Bx, in floc and liquor respectively; Al-Bireh: $9.24 + 1.80$, $8.61 + 2.07$ for Bd; $5.26 \pm 1.21$, $15.04 \pm 3.05$ for Bx, in floc and liquor respectively; Langenreichenbach: $6.67 \pm 0.89$, $6.71 \pm 1.52$ for Bd; $5.48 \pm 1.02$, $6.24 \pm 1.08$ for Bx; in floc and liquor respectively), creating local "communities" of potential predator–prey interactions of various sizes, including large hubs (Fig. 5).

Network structure differed between WWTPs with SH supporting a lower fraction of significant Kendall interactions, yielding the sparsest networks (Fig. 5, Table S3). Also, at this WWTP, more connections were observed in the flocs than in the liquor, in contrast to the other WWTPs (Table S3). As expected from the segregation observed between the bacterial communities in floc and liquor[22], prey OTUs common to the two microhabitats were infrequent (12.1 to 14.7%, Supplementary Data 2), and they were preyed upon by different predators; Identical predator–prey pairs common to flocs and the liquor were extremely rare as were predators preying on unique prey in a WWTP (Supplementary Data 2). In contrast, almost half of the predator OTUs were shared between the flocs and the liquor (41.2 to 48.8%, Table S3) but these predators were connected to different prey (Supplementary Data 2).

**Potential BALO prey ranges.** As predators exhibited potential wide to narrow prey ranges, as indicated by the number of connections between a predator node and different prey nodes (Fig. 5; Tables S4, S5), we posited this may affect their abundance. However, these parameters were not correlated ($r^2 = 0.001$ for both Bd and Bx). The sensitivity of a particular prey taxon to predation was also not necessarily linked to its abundance:

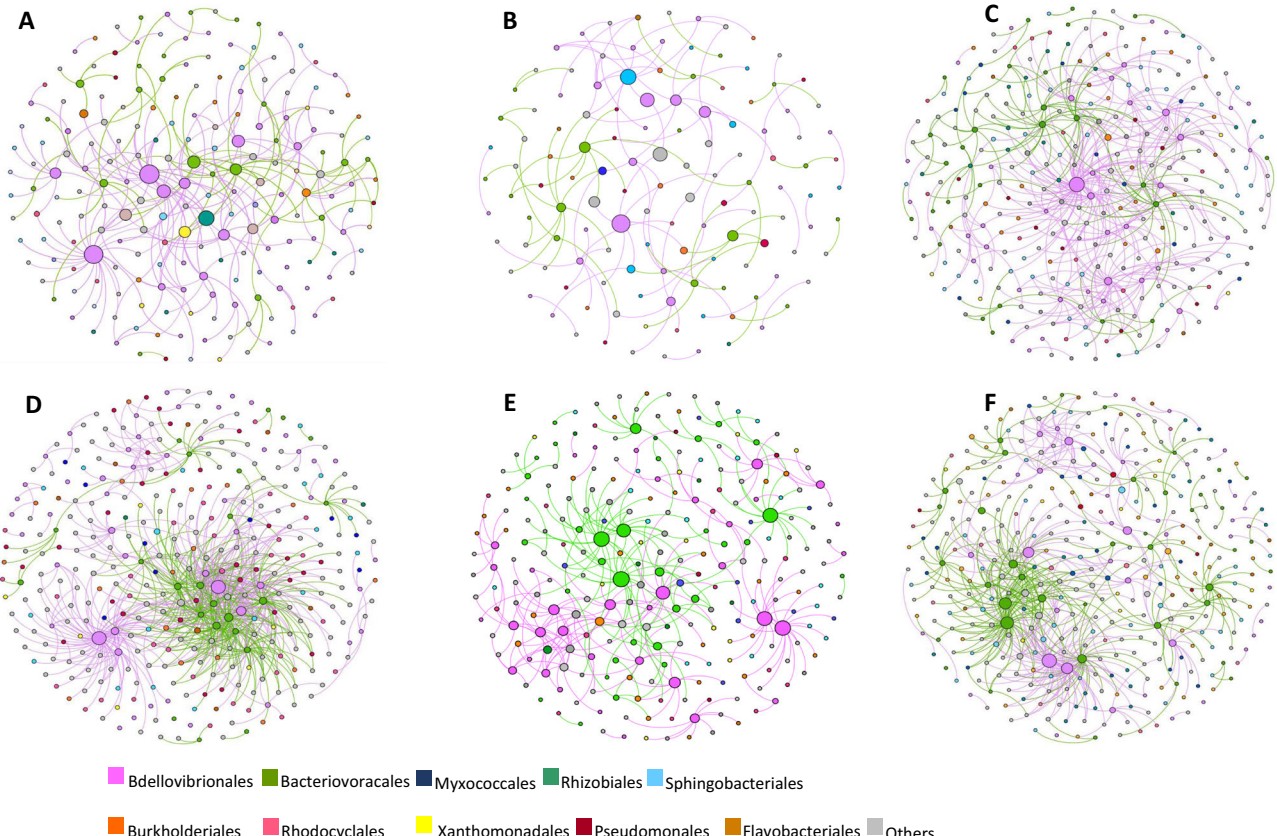

**Fig. 5 Network analysis of BALO predator–prey interactions.** At the Shafdan (**A**, **B**), Al-Bireh (**C**, **D**) and Langenreichenbach (**E**, **F**) WWTPs in the floc (**A**, **C**, **E**) and liquor (**B**, **D**, **F**) fractions. Connections stand for strong negative (Kendall τ < -0.7) and significant (*p* < 0.05) correlations. The size of each node is proportional to the number of connections (i.e. the number of prey OTUs that the predator is connected to).

Sphingobacteriales are the most abundant taxon (Figure S6) in these WWTPs and in WWTPs at large[28]. While very dominating in flocs (Figure S6A, C), they were proportionally more preyed upon in the liquor fraction (Figure S6D–F). Lower abundance taxa (<3%) like Myxococcales and Rhizobiales (Figure S6) could contribute a rather important fraction of the predator–prey edges (5.8 ± 3.8% and 3.1 ± 2.4%, respectively), while clades like Rhodocyclales appeared to be proportionally less preyed upon than their relative abundance would tell (4.9 ± 2.33% predation for a relative abundance of 11.7 ± 3.8%). Yet, at the prey OTU level, correlation analysis suggested BALO predation to be density-dependent (Fig. 4).

Although the same higher taxa (order level) were preyed upon by both Bd and Bx, the high modularity of the networks and their structures suggested differentiation in prey use by the different predators. Indeed, within each predatory family, different predators largely hunted different prey (47.7 and 50.9% of prey OTUs unique to specific Bd and Bx OTUs, respectively). We further examined prey resource partitioning between Bd and Bx at each site and within each fraction, observing that here also, the predators largely differed in potential prey range (Fig. 6).

**In situ identification of predator–prey interactions.** Growth experiments with different BALOs, including two strains expressing a fluorescent reporter protein were performed to confirm that BALOs indeed use WWTPs species detected in the network and hitherto not known to be BALO prey (Figure S7). Yet, this validation only includes a small fraction of the detected interactions. Thus, in order to validate the findings of the network analysis, an experimental design allowing specific sorting and analysis of interacting predator–prey populations, based on

in situ hybridization in conjunction with flow cytometry was developed. We first showed that using flow cytometry and a synchronous culture of a single predator and a single prey, bdelloplast (invaded prey) formation and the release of progeny free swimming attack phase (AP) predators can be quantitatively distinguished (Figure S8). Then, discrimination was achieved by forward (FSC) and side (SSC) scatter measurements, and by FSC and fluorescence after hybridization of the sample with *a Bdellovibrio* spp. targeted BDE525-Cy5' 16S rRNA nucleotide probe[23] (Figure S9). As a control, the antisense probe NonBDE525-Cy5' yielded almost no signals (Figure S9C, F). Following this, a sample of activated sludge was spiked with varying concentrations of mixed AP cells and bdelloplasts hybridized with the BDE525-Cy5' probe, showing that AP cells and bdelloplasts diluted to 0.1% of the total DAPI-counterstained cell population were detected (Figure S10).

Activated sludge samples from a plant neighbouring LB were then hybridized with a BDE525-Cy5' probe and stained with DAPI. Three different regions were identified and gated using FSC and Cy5' fluorescence: P1, representing a dense cloud of events; P14, a region surrounding P1 but of lesser density and intensity and; P16, a region with no positive Cy5' signals (Fig. 7). Using these gates, 150,000 cells from P1, 150,000 cells from P14, and 500,000 cells from P16 were sorted, out of a pool of $10^7.ml^{-1}$ cells, as measured based on DAPI staining. In order to validate that sorting resulted in the separation of single cells and not from aggregates or larger autofluorescent double cells, the pulse width was calculated. Accordingly, nearly all events were in the higher pulse width channels (P1, 98.5%; P14, 94%: P16, 95.9%) confirming single bacterial cells were measured during sorting (Figure S11). The NonBDE525-Cy5' control yielded very low

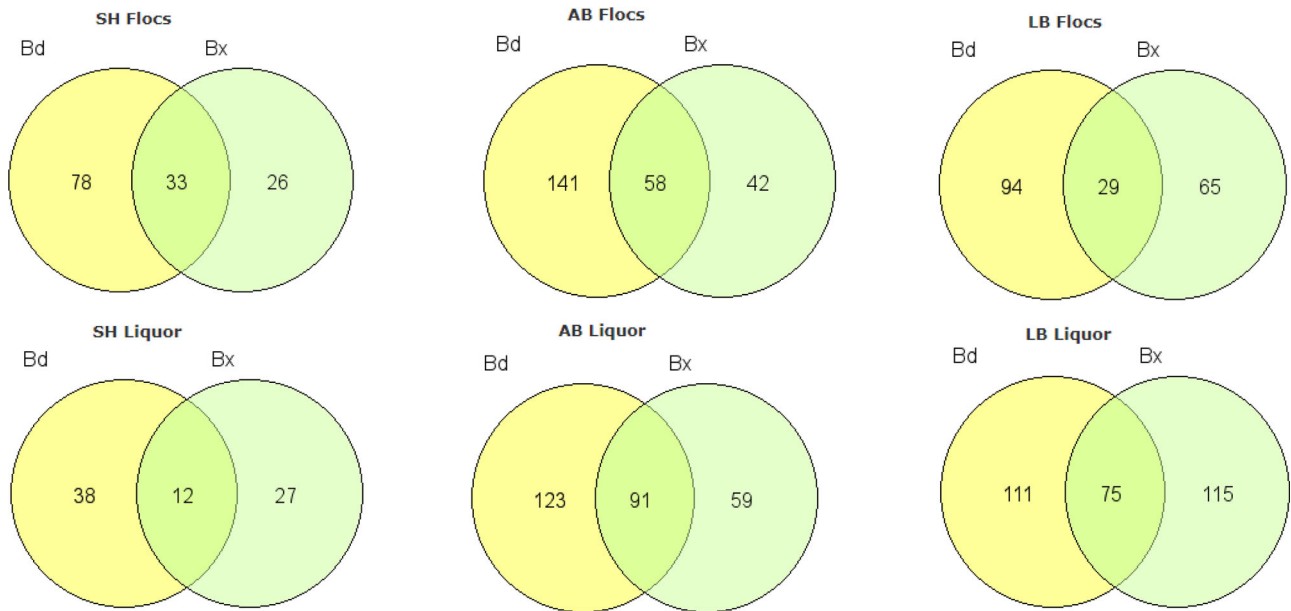

**Fig. 6 BALO prey ranges.** Prey range overlap (shared OTUs) between Bdellovibrionales (Bd) and Bacteriovoracales (Bx) in the floc and in the liquor fractions at the Shafdan (SH), Al-Bireh (AB), and Langenreichenbach (LB) WWTPs.

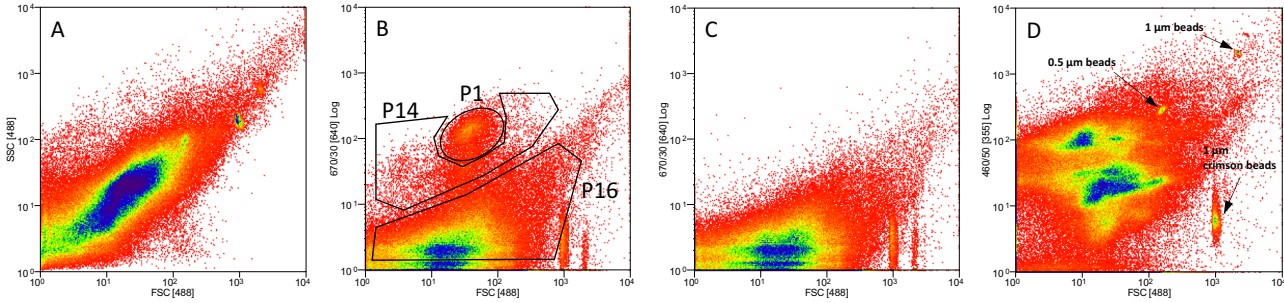

**Fig. 7 Flow cytometry (FC) analysis.** FC of a fixed activated sludge (AS) sample hybridized with a Bdellovibrio-specific BDE525-Cy-5' probe. **A** In the forward scatter (FSC)/side scatter (SSC) detection mode. **B** In the FSC/fluorescence (670/30[640]) (Fl) mode. Gates P1 and P14 define regions containing events positive for the BDE525 fluorescent signal. Gate P16 is signal-negative; **C** Fixed AS sample hybridized with a NonBDE525 probe, in the FSC/Fl mode, and; **D** Fixed AS sample hybridized with a *Bdellovibrio*-specific BDE525 probe, counter-stained with DAPI, in the FSC/fluorescence (460/50[356]) mode. Beads were used for the calibration and quantification of predator signals.

levels of signals (two orders of magnitude below that of BDE525) in P1 as well as in P14 (one order of magnitude). Further control for spurious attachment of BALOs to non-prey vs prey strains isolated from WWTPs by BALOs confirmed specificity (Figure S12).

DNA was obtained from the sorted samples using multiple displacement amplification (MDA). A PCR targeting the general bacterial population yielded a positive signal in all the samples, while a PCR using *Bdellovibrio*-targeted 16S rRNA primers yielded signals in samples sorted from the P1 and P14 gates only (Figure S13). In gate P16, the concentration of *Bdellovibrio* AP cells or *Bdellovibrio*-associated prey cells was mostly below detection level, thus validating the efficiency of the sorting process (Figure S13A). High-throughput 16S rRNA gene sequencing of the P1, P14 and P16 samples using universal 16S rRNA-gene primers, revealed depletion for gram-positive taxa in samples P1 and P14 at relative read abundances of $< 6.10^{-6}$ and $< 3.10^{-4}$, respectively, compared to about $6.10^{-2}$ in the P16 sample, providing further support that predators and the attacked prey (gram-negative) were very largely enriched in the BALO-sorted samples (Fig. 8A, $p < 0.01$). Principal coordinate analysis (PCoA) also showed that the bacterial

communities in samples P1 and P14 differed significantly ($p < 0.0005$) from the P16 community (Fig. 8B). This was reflected in their different compositions: while samples P1 and P14 mainly included sequences belonging to the family Bdellovibrionaceae, the families env.OPS_17, Saprospiraceae and Chitinophagaceae (Bacteroidetes), as well as Polyangiaceae (Myxococcales) and Campylobacteraceae, sample P16 was composed of Sphingobacteriales (with low abundance of env.OPS_17), Campylobacteraceae (mainly *Arcobacter* genus) and Chlorobiales. The total community was mainly composed of Rhodocyclaceae, Saprospiraceae, Comamonadaceae and Anaerolineaceae (Fig. 8C), a very similar composition to that of the nearby LB WWTP[22] from which the dynamics and network analyses described above were derived.

Finally, prey OTUs deduced from this FISH-cell sorting experiment were compared to those of the potential prey identified in the network data derived from the floc and liquor fractions at LB. More specifically, 64 (23.5%) and 46 (10.8%) of the OTUs co-sorting with BALOs, i.e. OTUs of the prey populations and identified in the P1 and P14 samples respectively, overlapped with potential prey OTUs of the LB networks (Supplementary Data 3).

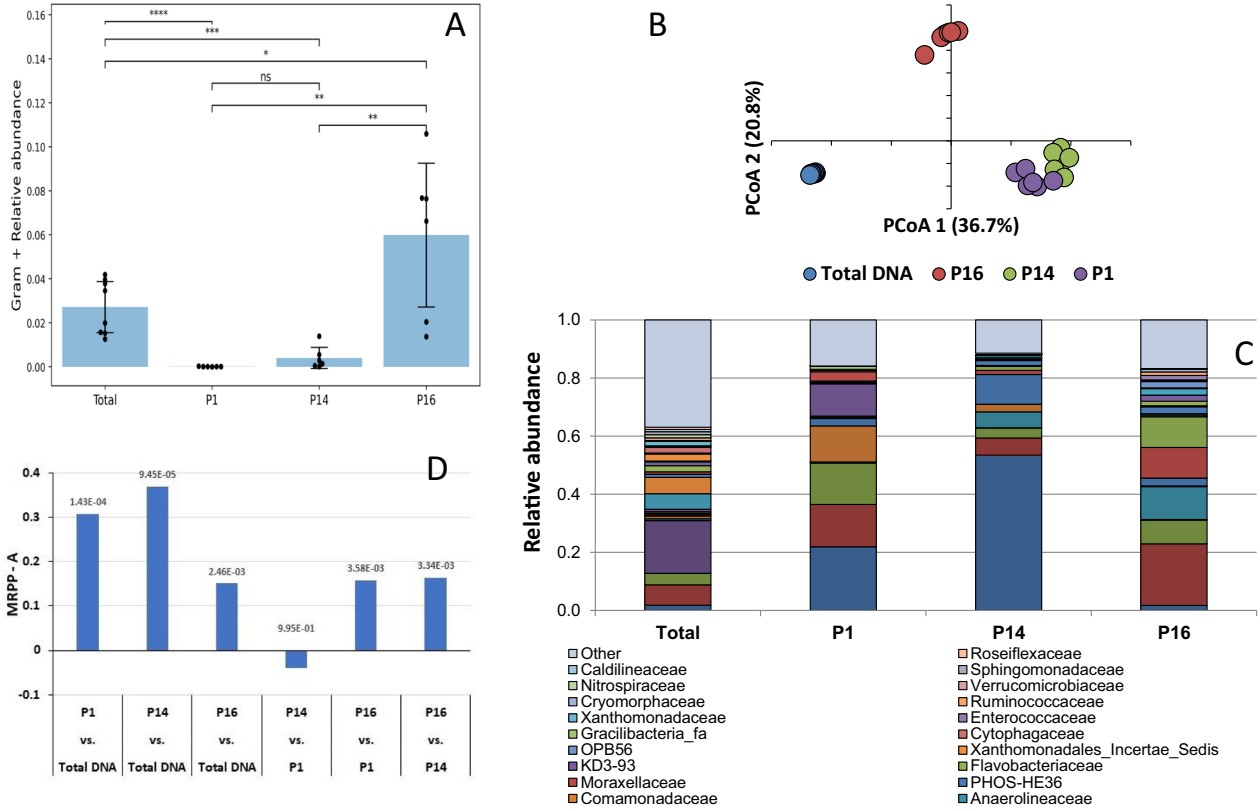

**Fig. 8 FACS and sorted populations sequencing analyses.** Gates P1, 14, and gate P16 define regions containing events positive and negative for the BDE525 fluorescent signal, respectively. **A** Relative abundance of gram-positive bacteria in the P1, P14, and P16 samples, and in total bacteria., determined from 16S rRNA gene community sequencing. Relative abundance was calculated by summing the abundance of reads assigned to the Firmicutes and Actinobacteria phyla, which contain the vast majority of gram-positive bacterial species divided by the total read number. Error bars show standard deviation Total DNA: 4 independent samples, in two technical replicates; All three gates were sorted twice, from three independent samples. Dots represent individual measurements. **B** Principal coordinate analysis of the sorted cells from the different gates. **C** Community composition analysis of the samples collected from gates P1, P14 and P16. Taxa are represented at the family level. **D** MRPP comparisons of Bray–Curtis distances between the communities in the sorted gates (P1, P14, P16) and the general community (total DNA). ns: $5.00e{-}02 < p <\, = 1.00e + 00$; *: $1.00e{-}02 < p <\, = 5.00e{-}02$, **: $1.00e{-}03 < p <\, = 1.00e{-}02$; ***: $1.00e{-}04 < p <\, = 1.00e{-}03$; ****: $p <\, = 1.00e{-}04$; no adjustment for multiple comparisons.

Among those, Saprospiraceae (12, 18.4%) were the most abundant, followed by Comamonadaceae (9, 13.8%). Chitinophagaceae (5, 7.7%), Nannocystaceae (4, 6.2%), Rhodocyclaceae (4, 6.2%), and Flavobacteriaceae (3, 4.6%) (Supplementary Data 3).

## Discussion

The large-scale BALO predators-targeted time series analysis performed here uncovers features of complex trophic networks linked to the activity of BALOs. It reveals a hitherto unknown diversity of the predators; exposes the highly dynamic nature of the BALO community; uncovers differential temporal dependence of BALOs upon seasons in the long term and possibly upon prey density on shorter time frames, and; appears to explain the co-existence of numerous predators by prey range partitioning. It also shows that BALO communities are geographically largely distributed, exhibit distance decay, and appear to be subjected to local, ecological selection. However, and in contrast to the general bacterial community[22], BALOs are not spatially segregated in the connected microhabitats forming the active fraction of WWTPs. Finally, the study establishes an approach that enables tracking and identification of interacting predator and prey in a complex system and an experimental approach for validating sequence-based interaction networks.

The high BALO diversity observed included highly dominant BALO OTUs present in almost all samples in the three studied WWTPs, as well as rare ones. Almost all OTUs differed from known, cultured strains. Only one of the dominant Bdellovibrionales OTU (Bd03) diverged by no more than 3% from described strains, including *Bdellovibrio* W, a rare cyst forming strain[29]. Few OTUs clustered near strain JSS, an epibiotic predator isolated from WWTP[10]. Unless epibiotic predators are to be found in uncharacterised clades, this suggests that epibiotic predation is uncommon. In the Bacteriovoracales, only a few, non-dominant OTUs were similar to the terrestrial *Bacteriovorax stolpii* strain, all the other OTUs being more distantly related to described isolates. Thus, isolated BALOs are biased towards specific clades, representing but a very small part of Bd and Bx diversity. So far, most prey used for BALO isolation belong to the β– and to the γ–Proteobacteria, but this study shows that at least in WWTPs, Sphingobacteriales, α–, δ–Proteobacteria, and Rhodocyclales appear to form the majority of the prey. Recent large-scale metagenomics studies reveal that up to 5% novel bacterial taxa are missed from 16S rRNA gene community analyses in WW[30,31]. This may include unknown obligate predators and prey and thus further increase predator and prey diversity. Even "standard" prey like *Escherichia coli*, *Pseudomonas* spp., *Vibrio parahaemolyticus* used to isolate marine BALOs[8,32], may therefore only yield snapshots of the genuine diversity of the predators.

The BALO predatory communities, similar to the general microbial (bacterial and micro-eukaryotic) community[22] appeared to experience distance decay, as they were more similar at SH and AB than between these two relatively close by plants, and LB. However, for both Bd and Bx, the 20 most abundant OTUs were almost all shared between the plants, while rarer OTUs were much less common. This is suggestive of no or few constraints on dispersal as observed with other bacterial WWTP populations[22], and of local selection for the less abundant OTUs. Such a pattern was observed in Chinese lakes in which the local environment selected for rare genotypes more strongly than for dominant genotypes, which in turn, were more influenced by regional conditions[33].

Resource partitioning can be invoked to explain the co-existence of the numerous predators. Partitioning can be sustained through prey range differences between predators, as shown by our network-based prey range analysis and by previous culture-based studies[11,12] or by spatial separation of the predators. However, flocs and liquor microhabitats did not segregate BALOs, as the Bd and the Bx communities did not differ between the floc and the liquor microhabitats. This is in contrast to the micro-eukaryote predators and the general bacterial community[22]. It may yet be that biofilms on constructed surfaces or on biological surfaces (e.g. rotifers, daphnias, nematodes, or fungi) provide additional microhabitats for BALOs and prey to interact. Yet, few of the prey OTU (circa 6%) detected by network analysis were shared between flocs and liquor. BALO predation is thus not impaired by prey strain specificities promoting floc colonization[34,35], in accordance with previous data showing that BALOs are effective predators of planktonic as well as of biofilm-associated cells[20,36–38]. The lack of differences in BALO distribution between flocs and liquor may be due to their ability to move between these microhabitats: BALOs infiltrate sludge bacterial biofilms[14,20] prey upon planktonic cells[39], floc surface bacteria, and enter the inner parts of a floc where oxygen can penetrate[40]. They also do not segregate into surface-associated and planktonic types. The propensity of floc surface-associated bacteria (including BALO-infected ones), to be washed out to the liquor fraction[40,41] may further contribute to minimize population segregation between the microhabitats. This, in spite of flocs supporting a larger population of BALOs, a consequence of the higher bacterial density in the floc fraction, and hence of potential prey, than the liquor. Accordingly, BALOs may not impose significant differential selection between floc and liquor-associated prey, which can lead to the formation of predation-resistant floc-associated bacteria, as observed with protists[42].

The total community size of the predators was relatively stable, fluctuating between 0.1 and 1% of the total bacterial community, suggesting that the carrying capacity of the predators is strongly linked to that of the total community. However, BALO OTUs largely fluctuated during the time series, succeeding one another, showing oscillation-like patterns. These oscillations could differ between flocs and liquor, possibly due to subtle effects such as differences in prey susceptibility between the planktonic phase and biofilms[36,43] or even between closely related strains[12,44]. BALO OTUs oscillated at frequencies spanning from weeks to season (winter/summer), a feature observed at the three WWTPs providing yet another mechanism to further sustain co-existence of the diverse BALO populations through temporal differentiation[45,46]. Yet, the main drivers of these two types of temporal shifts may be different: season-long oscillations in WWTPs appeared to correlate with temperature, a parameter that can directly, or indirectly through the alteration of prey abundance, affect the BALOs[9,16,47]. Shorter-term oscillations may be driven by other mechanisms: "Kill the Winner" (KtW) models are based on the assumption that the fitness of an organism decreases

as its frequency (relative abundance) increases, leading to predation being the major regulatory mechanism for community composition under highly productive environments[48] like WWTPs. Experimental support for KtW in WWTPs was provided by Shapiro et al.[49] who showed predator (phage)-prey patterns consistent with KtW. BALO populations also fluctuate in abundance according to the presence of prey, and they can do so rapidly: Specific BALOs are indeed rapidly selected for by an increase in the abundance of specific prey[9,13]. Our network analyses based on negative Kendall correlations showed KtW-like patterns with multiple oscillations. Although oscillations in the present study were at the same time scale as those observed with phages in the Shapiro et al. study[49] (tens of days to a few months), some were more rapid. Yet, not all strongly negative correlations supported clear KtW patterns, possibly because in addition to prey availability, other biotic and abiotic factors, like responses to prey depletion or the secretion of predation-inhibitory substances[50–52], and environmental parameters (see below) may affect predatory dynamics. In both cases, no relation with prey relative abundance in the WWTP was observed as both KtW and non-KtW patterns could be seen with less (Rhizobia, Flavobacteria) and more abundant (Sphingobacteria) prey. The networks and the detailed prey-range analyses showed large numbers of connections between BALOs and orders like Rhizobiales, Xanthomonadales and Flavobacteriales, which are found at a rather low abundance, seemingly contradicting a KtW approach. Yet, KtW has been extended to include more realistic situations than when first proposed, by taking into account the selective grazing of prey by protozoa resulting in different population groups, and interactions of multiple predators with multiple prey[53]. BALOs similarly have differential access to prey as they have different prey ranges that potentially vary from restricted to generalist[11,54,55], represented as large "hubs" in the networks, which in turn, are thought to indicate overall community stability[56].

Interestingly, relatively more Bd OTUs (105 of 133 OTUs) mapped upon the predator–prey networks than Bx OTUs (94 of 537 OTUs), suggesting that environmental factors may also directly affect BALO dynamics. Although unmeasured variables may contribute to about 50% of the variance in WWTP sludge communities[57,58], some studies showed the effect of particular environmental variables on BALOs. For example, selection for discrete phylogenetic Bx clusters along a salinity gradient in the Chesapeake Bay provides such an example of a strongly selective environmental parameter[21]. More recently, Welsh et al.[25] using network analysis showed that temperature and nutrients changed *Halobacteriovorax* (Bacteriovoracales)-prey interactions in a coral microbiome. Damping of BALO-prey interaction signals resulting in their absence from the networks may, in addition to abiotic factors, be caused by direct predation upon BALOs by phages[59–61], by generalist, top predators like protists that prey upon the bacterial predators as well as upon their prey[62] or by competition between predators. Such may be the case with Myxococcales, a clade of facultative predators[63] that while efficiently preying upon gram-positive prey[64] can also utilize gram-negative prey and potentially compete with BALOs. We found Myxococcales to be highly connected to BALOs, representing up to 4.4% of total edges, suggesting they are largely preyed upon. This corroborates previous findings[25] in which network analysis showed the potential removal of these predatory competitors by BALOs. These additional interactions between predators reflect complex intraguild predation relationships, which may have important repercussions upon the trophic network by affecting predator diversity, lowering prey suppression rates[5] and indirectly constraining the evolution of predator–prey interactions[65].

Sphingobacteriales, a dominant group in WWTPs[22], including Saprospiraceae and Chitinophagaceae was largely preyed upon (10.4 %). Community analysis of BALO-spiked WW samples showed predation upon these taxa, a result further supported by predation experiments on strains isolated from WWTPs[38,40,66,67]. Sphingobacteria hydrolyse proteins and various organic compound[68], likely living off dead biomass, EPS or other soluble microbial products. They are amongst the most dominant bacteria in WWTPs (16–70% of the floc fraction, 3–32% in the liquor fraction[22]; 10.1–15.8% in activated sludge[69]). They play a crucial role in sludge autolysis and reduction of sludge biomass[69], which are highly important to WWTP operations, reducing sludge disposal costs. Predation by BALOs may also affect other important functional groups like the denitrifying *Comamonas* (Comamonadaceae, totalling 6.8% of the edges). Noteworthy, a previous network analysis of WWTPs bacterial communities suggested that these taxa are linked to *Bdellovibrio*[70].

Although similar linkages between BALOs and prey have been obtained by network analysis in independent studies and in different environments, providing strong support that network analysis can accurately predict predatory interactions, so far no experimental confirmation of the validity of such networks has been performed. To address this issue, FISH-labelled cells were sorted from sludge. Cell sorting can be used to separate specific populations in wastewater in a high-throughput manner, after highlighting them with fluorescent dyes[71] or directly by selectively sorting fluorescently labelled target populations using directed FISH probes[72–74]. Here, we employed this approach to sort interacting predators and prey to obtain direct identification of these populations. So far, in situ identification in environmental samples has been performed with stable isotopes labelling prey or growth substrate, leading to the identification of a succession of *Halobacteriovorax* 16S rRNA sequences in a tidal cycle[15] and of a *Micavibrio* predator interacting with a sublineage type I *Nitrospira* in a WWTP[19].

A number of controls were performed to assure the validity of the analysis: prey and non-prey strains were tested against a BALO predator to validate attachment to prey but not to non-prey; an anti-sense *Bdellovibrio*-targeted probe yielded no signal when hybridized to sludge samples; pulse width measurements confirmed that the gated regions were constituted of single cells; *Bdellovibrio* spiked into sludge samples were detectable down to low densities (0.1% of the total cell numbers, equivalent to the lowest densities of BALOs in our samples); very few sequences affiliated to gram-positive taxa were detected in the sorted predator fractions, and; sequences affiliated to *Bdellovibrio* were only detected in the sorted predator fractions. The data revealed that *Bdellovibrio* contributed 12.1% of the sequences in the P1 samples, and only 3% in the P14 sample, a 12-fold and 3-fold enrichment, respectively, as compared to the control sample. The P14 sample also included events of relative lower signal intensity, contributing to the lower relative abundance of predators. It can be noted that if every sorted event originated from an AP cell (thus a predator alone) or from a bdelloplast (a predator associated with a prey) one could have expected at least 50% *Bdellovibrio* sequences in the sorted samples. However, BALOs' genomes hold one to two rRNA gene copies[75] while the largest prey populations, i.e. the Saprospiraceae, Chitinophagaceae, various Myxococcales, Pseudomonadales and Comamonadaceae contain between one to six, two to four copies and up to nine rRNA gene copies per genome, respectively (https://rrndb.umms.med.umich.edu/), possibly explaining part of the discrepancy. Remarkably, the samples used in this analysis originated from another WWTP situated near the LB WWTP, which provided the samples upon which the LB network was constructed. Yet, samples originating from both were composed of

the same taxa, and included a significant fraction of identical prey OTUs. Lastly, our FISH-FACS experiment employed different DNA extraction/PCR approaches than the one used to obtain sequences for community analysis. Yet, they support one another. Taken singly and together the results strongly suggest that our network approach is valid, providing a powerful tool for in situ tracking of predator and prey dynamics in complex natural, as well as in controlled, constructed environments.

This approach may be developed to precisely identify interacting individual predator and prey, for instance by using specific probes for each, based on computed interactions or by isolating single predator–prey pairs (bdelloplast) and performing high-throughput single-cell sequencing or by directly linking interacting pairs using PCR technologies such as epicPCR[76].

BALOs have been proposed to form a "sideways" control over bacterial succession[15]. So far, their relative importance in biomass control and bacterial turnover have been largely underappreciated, for instance as predators of bacteria protected from protistan grazing, which can be very abundant in WWTPs and in freshwater[7,77]. Furthermore, numerous recent studies show that BALO abundance can be greatly responsive to the type of WWT technology[78], offering novel possibilities in WW treatment. Our cell sorting approach although still limited by methodological constrains like DNA extractability[79] and the number of retrieved cells, fills a crucial knowledge gap and provides the mechanistic details needed to quantify BALO-imposed bacterial mortality, and their contribution to microbial turnover in trophic networks.

## Methods

**Sampling locations and procedures.** Sampling was performed at the WWTPs of Shafdan (Israel), Al-Bireh (Palestine) and Langenreichenbach (Germany). Samples were taken monthly for a year from March 2013 to February 2014 and additionally, for four consecutive weeks in August 2013 and in February 2014. Overall characteristics of the sites are described in Table S4 (A, B).

One-litre samples retrieved from the activated sludge basin were transferred on ice to the lab within hours and further subjected to chemical analyses. The liquid (bulk water, supernatant) fraction of the activated sludge was separated from its floc fraction by sedimentation on ice for 1 h. Both the liquid and the sludge fractions summed up to 198 samples that were further processed as in Cohen et al. [22] and analysed with 16S rRNA gene amplicon sequencing targeting the Bdellovibrionales and the Bacteriovoracales. In addition, samples were taken from activated sludge basins from Eilenburg, Germany situated about 20 Km from Langenreichenbach. Two hundred millilitres of samples were taken and further fixed, processed and subjected to BALO-specific FISH-labelling, flow cytometric analysis and cell sorting.

**High-throughput 16S rRNA gene sequence analysis.** MiSeq Illumina sequencing (Illumina, Carlsbad, CA, USA) was performed as previously described[80]. DNA from samples from the Shafdan, Al-Bireh and Langenreichenbach WWTPs were amplified using the 16S rRNA gene primers of Bdellovibrionales, Bd824F (5'-ACTTGTTGTTG GAGGTAT-3')-Bd1222R (5'-TTGTAGCACGTGTGTAG-3'), and of Bacteriovoracales, Bx341F (5'-CTACGGGAGGCAGCAG-3')-Bx672RC (5'-TACCCCTACATGCGAAA TTCC-3') (Table S5). DNA from samples originating from Eilenburg, Germany, WWTP and used in the FISH labelling experiment (see below) were amplified using 16S rRNA gene primers targeting: total bacteria, 515F (5'-GTGCCAGCMGCCGCGG TAA-3')-926R (5'- CCGYCAATTYMTTTRAGTTT-3') (Walters et al. 2015), and; the Bdellovibrionales, as above.

Sequences were processed using MOTHUR v1.4[81]. FASTA and quality data were first extracted from the raw FASTQ file. Sequences were grouped according to barcode and primer, allowing one mismatch to the barcode and two mismatches to the primer. Quality control, trimming and de-noising were performed as outlined in the standard MOTHUR MiSeq protocol (http://www.mothur.org/wiki/MiSeq_SOP). All sequences were aligned to the SILVA v. 132 reference alignment database[82] and filtered, so that they all overlap perfectly (with no overhang). To further reduce sequencing errors, sequences were pre-clustered based on the algorithm of Huse et al. [83]. Finally, chimeric reads were removed with MOTHUR's implementation of the UCHIME method[84], and all chloroplast, mitochondria, and 'unknown' (i.e. unclassified at the kingdom level) reads were deleted. In total, 7.96 million reads with a uniform length of 291 nucleotides per read for the 16S rDNA amplicons, 8.1 million reads of Bdellovibrionales with a uniform length of 390 bp per read and 4.1 million reads with a uniform length of 300 bp per read for the Bacteriovoracales amplicons, averaging 35633 ± 10244, 39002 ± 20794, and 20207 ± 12391 reads per sample, respectively, were obtained from the Shafdan, Al-Bireh and Langenreichenbach WWTPs. In the FISH-FACS experiments, 879,042 reads using the general bacteria primers and 633,852 reads using the primers for

Bdellovibrionales 16S rDNA were obtained, for 32613 ± 6142 and 24,378 ± 6224 reads per sample, respectively. Pairwise distances were calculated between all DNA reads, and reads were subsequently binned into operational taxonomic units (OTUs) at the 0.03 level (> 97% similarity). The taxonomic affiliation of each OTU was based on the current SILVA v.132 taxonomy[85].

**Microbial diversity and statistical analyses**. The OTUs were arranged in a data matrix where each row was a single sample and each column a specific OTU; each data point in the matrix represented the abundance of the particular OTU in a particular sample, relativized to the sampling effort (i.e. the number of MiSeq reads obtained from that sample). Rarefaction curves were calculated using MOTHUR Miseq protocol (http://www.mothur.org/wiki/MiSeq_SOP). Read abundance data were not rarefied[86]. α-diversity parameters (Shannon, Simpson, Richness and Evenness) and multivariate analysis were performed in PC-ORD v6.0 (MjM Software, Gleneden Beach, OR) with Sorensen (Bray–Curtis) distances. Ordinations were performed with non-metric multidimensional scaling (NMDS)[87] at 500 iterations. Differences between sample groups were calculated with the multi-response permutation procedure (MRPP)[88] a test based on the assumption that if two groups differ, the average within-group difference is smaller than the average between-group distance. The size of the difference between groups was represented by the A-statistic of the MRPP test, while its significance was identified by the MRPP P-value. Correlations between the microbial communities (as represented by the PCoA 1st axis) and the environmental parameters were calculated using Pearson's correlation coefficient.

**BALOs phylogenetic analysis**. BALOs sequences were aligned using MUSCLE (Multiple Sequence Comparison by Log-Expectation)[89]. Gaps and poorly aligned positions were eliminated. The resulting unambiguously aligned 390 base pair sequences of Bdellovibrionales and 300 base pair sequences of Bacteriovoracales were reconstructed into a maximum likelihood tree using MEGA6[90] with known Bdellovibrionales and Bacteriovoracales isolate as the outgroup. The bootstrap consensus tree was inferred from 500 replicates. Branches corresponding to partitions reproduced in < 50% bootstrap replicates were collapsed. The phylogenetic tree was visualized using the programme iTol v4 (https://itol.embl.de/).

**Co-occurrence network analysis**. All three annual datasets (total bacteria, Bdellovibrionales and Bacteriovoracales) were grouped together for each WWTP and each fraction (floc/liquor). All possible Kendall rank correlations[91] between all bacteria[22] and all bacterial predator OTUs which had an abundance of more than ten reads in each WWTP and in each fraction were calculated. This threshold was applied in order to filter poorly represented OTUs and reduce network complexity. Between 3374 and 3833 bacterial OTUs from our previous study[22] were correlated against 99–114 Bdellovibrionales OTUs and 59–81 Bacteriovoracales OTUs. A correlation was considered robust when the Kendall correlation coefficient (τ) was both < -0.7 and statistically significant (P-value < 0.01). The nodes in the constructed network represent the OTUs at 97% identity according to the taxonomy derived from the SILVA taxonomy. The topology of each network was described according to a set of measurements (average node connectivity, average path length, diameter, cumulative degree distribution, clustering coefficient and modularity)[27]. Overlaps between the potential prey ranges were calculated using GenEvenn (http://genevenn.sourceforge.net/index.htm). Correlations and statistical analysis were carried out with SciPy package of Pandas 0.23.4 under an ipython platform and networks were calculated, explored and visualized using the interactive platform Gephi v0.9.2i[92].

**Bacterial strains and growth conditions**. B. bacteriovorus strain HD100 was grown in DNB medium amended with 3 mM MgCl2 and 2 mM CaCl2 together with Escherichia coli strain ML-35 prey cells for 24 h at 28 °C with vigorous aeration, yielding about $10^9$ B. bacteriovorus attack phase (AP) cells.ml$^{-1}$. When needed, AP cells were filtered through a 0.45μm membrane to remove remaining prey cells, rinsed by centrifugation at 10,000 g, 10 min followed by resuspension in 5 ml of amended HEPES buffer (25 mM HEPES [Sigma-Aldrich, Germany], pH 7.8 with 3 mM MgCl2 and 2 mM CaCl2). Growth phase (GP) B. bacteriovorus cells were obtained by mixing AP cells and prey cells at a 1:10 ratio in amended HEPES, creating bdelloplasts (infected prey cells) within which the predatory cell grows. B. bacteriovorus HD100 and 109 J and 109 J carrying the tdtomato reporter gene[93] were incubated for up to 280 h in microtiter plates to test for attachment and predation of a variety of isolates from WWTPs (figures S7 and S12). For detailed protocols, see[94]

**Bacterial quantification by real-time qPCR**. Standards were prepared by inserting a 1467-bp fragment of the Bdellovibrio bacteriovorus HD100 and Bacteriovorax stolpii UKi2 strain 16S rRNA gene amplified with primers 27 F and 1492 R respectively into a PGEM-T easy plasmid vector system (Promega). Ten-times serial dilutions from $10^3$ to $10^{10}$ plasmid copies per reaction were used to construct standard qPCR curves and plasmid copy numbers calculated[95]. For total bacteria, primer pair 1048F-1175R[96] was used to quantify the 16S rDNA copy number. Each 25 μl reaction consisted of 12.5 μl of SYBR® Green PCR Master Mix (Applied Biosystems) 1 μl of each primer, 1 μl of DNA and 9.5 μl of PCR grade DDW.

Thermal cycling was performed as follows: 50 °C (2 min) and 95 °C (10 min), 40 cycles of 95 °C (15 sec), 60 °C (1 min), followed by dissociation; Melt curve was used for all experiments from 55 °C to 95 °C. For Bdellovibrio 16S rDNA quantification primers Bd347F-Bd549R[97] were used in a reaction mix and thermal cycling conditions as above except for the use of 45 cycles. For Bacteriovorax 16S rDNA quantification primers BacF519 - BacR677[98] were used in a reaction mix as above. Thermal cycling conditions were: 50 °C (2 min), 94 °C (2 min), 45 cycles at 94 °C (30 sec), 62 °C (10 sec) and 72 °C (10 sec), followed by dissociation; Melt curve was used for all experiments from 55 °C to 95 °C. Reactions were performed in a 96-well plate (Applied Biosystems) with MicroAmp® Optical Adhesive Film (Applied Biosystems) in a Step One plus Real-time PCR System (Applied Biosystems). Total QPCR counts (16S rRNA bacterial genes ml$^{-1}$) were estimated based on a standard curve with 100% efficiency factoring in the cycle threshold value of the particular sample obtained, taking into account that 25 mL of sample was used for DNA extraction, DNA was eluted in 50 μl of DDW and 1 μl of this DNA was 10 fold diluted and used in the QPCR reaction.

**Cell fixation and cell staining**. For flow cytometric analysis of the wastewater community, samples were treated as described in Liu Z, et al.[99]. In short: Samples were taken from the activated sludge basin of a wastewater treatment plant in Eilenburg, Germany, centrifuged (3,200 g, 10 min, 4 °C) and the supernatant discarded. The cells were washed with phosphate-buffered saline once (PBS, 6 mM Na2HPO4, 1.8 mM NaH2PO4, 145 mM NaCl, pH 7, 3,200 g, 10 min, 4 °C) and stabilized by adding 2 ml paraformaldehyde solution (PFA, 2% in PBS) to the cell pellet and incubated for 30 min at room temperature (RT). After another centrifugation step (3,200 g, 10 min, 4 °C), 4 mL of EtOH (70%) were added for fixation and the cell solution was stored at -20 °C. For DNA staining these samples were washed twice (3,200 g, 10 min, 4 °C) with PBS, and cell solutions were adjusted to an OD of 0.035 (d$_{\lambda 700nm}$ = 5 mm) with PBS. Two millilitres of an adjusted sample was centrifuged (3,200 g, 10 min, 4 °C), and the pellet resuspended with 1 mL solution A (0.11 M citric acid and 4.1 mM Tween 20, with distilled water) and incubated at RT for 10 min in an ultrasonication bath (Merck Eurolab, Darmstadt, Germany) and 10 min without any further treatment. After another centrifugation step (3,200 g, 10 min, 4 °C) the cells were stained with solution B [0.24 μM DAPI (4',6–diamidino-2-phenylindole) in phosphate buffer (289 mM Na2HPO4 and 128 mM NaH2PO4 in distil water)] overnight at RT.

For FISH experiments, AP and GP cells obtained from a dual culture of B. bacteriovorus HD100 and E. coli ML-35 were centrifuged at 10,000 g for 10 min and suspended in final 1% PFA for 2 h at room temperature. Fixed cells were then washed twice with 500 μl PBS, pelleted as above, suspended in 50% EtOH-PBS and stored until use at -20 °C. Samples were strongly vortexed before hybridization with FISH probes. One millilitres of fixed sample was diluted with 2 ml of PBS, vortexed again, and placed in a sonication bath for 5 min. Samples were then washed twice with PBS and sonicated again for 10 min. When required, samples were concentrated to 100 μl in PBS.

Hybridization was performed with a Bdellovibrionales 16S rRNA-targeting, 5'-Cy5 labelled BDE525 (5'-GATCCCTCGTCTTACCGC-3') probe[23]. Controls included target and non-target (E.coli ML-35) organisms to confirm probe specificity, and a general bacteria-directed FITC-labelled EUB338 (5'-GCTGC CTCCCGTAGGAGT-3') probe[100]. All experiments also included a NONBDE525 probe (5'-GCGGTAAGA CGAGGGATC-3'), an antisense probe to BDE525, as a negative control for unspecific binding. Fixed samples were mixed with hybridization buffer (HB, 5 M NaCl, 1 M Tris–HCl, 30% formamide, 10% SDS) and the appropriate probe (5 μl pure cultures, 47 μl HB, 3 μl of a 10 mM probe stock; 50 μl sludge, 470 μl HB, and 30 μl of a 10 mM probe stock) treated for 2 h at 46 °C in an hybridization oven. Samples were pelleted, washed with 100 μl wash buffer (5 M NaCl, 1 M Tris–HCl, 0.5 M EDTA, 10% SDS) at 48 °C for 20 min. Counter-staining with DAPI was performed by adding 445 μL of either PBS or DAPI-buffer (400 mM Na2HPO4/ NaH2PO4, pH 7.0) (Sigma-Aldrich) and 5 μL of a 50 μM DAPI solution to a 50 μL of BDE525-Cy5 labelled activated sludge sample. Samples were incubated at room temperature for 2.5 h up to overnight, and kept in the dark at 4 °C until use.

**Flow cytometry and cell sorting**. Cytometric measurements were performed with a BD Influx v7 Sorter USB, (Becton, Dickinson and Company, Franklin Lakes, USA) equipped with a 488 nm Sapphire OPS laser (400 mW), a 355 nm Genesis CX laser (100 mW, both from Coherent, Santa Clara, CA, USA), and a red 640 nm 56CRH laser (120 mW, Melles Griot, Carlsbad, CA, USA). The 488 nm laser light was used for the analysis of forward scatter (FSC, 488/10), side scatter (SSC, 488/10, trigger signal), and yellow–green fluorescence (PMT3, 540/30). Cy5 fluorescence was recorded at PMT7 (670/30) after excitation with 640 nm light. DAPI–fluorescence was measured at PMT9 (460/50) after excitation with 355 nm light.

The fluidic system ran at 33 psi using a 70 μm nozzle. The sheath fluid consisted of 0.5 x FACS flow buffer (BD). For daily optical calibration of the cytometer in the linear range, 1 μm blue fluorescent FluoSpheres (Molecular Probes, F-8815, Eugene, OR, USA) and 2 μm yellow–green fluorescent FluoSpheres (ThermoFisher Scientific, F8827, Waltham, MA, USA) were used. For calibration in the log range, 0.5 μm UV Fluoresbrite Microspheres (Polysciences, 18339, Warrington, PA, USA), 0.5 μm yellow–green fluorescent Fluoresbrite - Carboxylate Microspheres

(Polysciences, 15700, Warrington, PA, USA), and 0.2 μm crimson fluorescent Fluospheres-Carboxylate Microspheres (ThermoFisher Scientific, F-8806, Waltham, MA, USA) were applied. Prior to measurement, samples were spiked with 0.5 μm UV Fluoresbrite Microspheres (both Polysciences, 18339 and 17458, Warrington, PA, USA). The microspheres served as internal standards to monitor instrument stability and to allow the correct comparison of samples. Cell data were collected in logarithmically scaled 2D-dot plots according to fluorescence and FSC for cell size-related information. Gates were set for apparent cell clusters and the cell sorting was performed according to a standardized procedure described before[99]. Briefly, cell were sorted using the '1.0 Drop Pure' sort mode. Either 150,000 cells per gate (P1 and P14) were sorted at a sort rate of about 80 cells.sec$^{-1}$ or 500,000 cells (P16) were sorted at a sort rate of about 1,800 cells.sec$^{-1}$ into tubes. Sorted cells were harvested by two successive centrifugation steps (20,000 g, 6 °C, 20 min), and the cell pellets were stored at − 20 °C for subsequent downstream analysis.

To compare *B, bacteriovorus* attachment to prey and non-prey, tested strains were incubated with the predator for 24 h[94]. Samples taken at 0, 2, 4, 6 and 24 were fixed and measured by flow cytometry using a 355 nm UV-Line for excitation[101]. All measurements were made comparable to each other by spiking 1 μm and 0.5 μm calibration beads in each of the samples. All flow cytometric raw data files are deposited in the FlowRepository database (www.flowrepository.org) with the repository ID: FR-FCM-Z3WS.

**Multiple displacement amplification (MDA) and PCR validation**. Sorted cells from the three gates (P1, P14: 150,000 each; P16: 500,000) were transferred to Eppendorf tubes (~ 120 μl per sample), pelleted at 20,000 g for 20 min at 4 °C, and the supernatant removed. DNA MDA was performed using the Illustra GenomiPhi V2 DNA Amplification Kit$^{TM}$ (General Electric, Germany) according to the manufacturer's protocol. Briefly, 1 μl of suspended cells was mixed with 9 μl of sample buffer, denatured at 95 °C for 3 min, cooled on ice, added to 9 μl of reaction buffer and 1 μl of Phi29 DNA polymerase. The reaction was incubated at 30 °C for 1.5 h and then inactivated at 65 °C for 10 min.

The presence of members belonging to the Bdellovibrionales was validated using the specific 16S rRNA gene primers Bd824F ('5-ACTTGTTGTTGGAGGTAT-3') and Bd1222R ('5-TTGTAGCACGTGTGTAG-3') targeting the Bdellovibrionales and polymerase chain reaction amplification as follows: 95 °C – 5 min, 95 °C -30 sec, 48 °C -45 sec, 72 °C -30 sec, for 28 cycles; and 72 °C -7 min. The presence of general bacteria and other controls were validated using primers 515 F ('5-GTGCCAGCMGCCGCGG TAA-3') and 806 R ('5-GGACTACHVGGGTWTCTAAT-3') and the following PCR program: 95 °C—5 min, 95 °C—30 sec, 55 °C—45 sec, 72 °C—30 sec, for 28 cycles; and 72 °C—7 min. PCR products were run on 1.2% agarose gels at 70 V for 35 min and photographed using a UV illuminator.

**Reporting Summary**. Further information on research design is available in the Nature Research Reporting Summary linked to this article.

## Data availability
The sequencing data generated in this study have been deposited under the associated BioProject, SRA, and BioSample accession numbers are at https://www.ncbi.nlm.nih.gov/bioproject/PRJNA715957. All row flow cytometry data are deposited in the FlowRepository database (www.Flowrepository.org) under the repository ID: FR-FCM-Z3WS. Source data are available in the supplementary material. Source data are provided with this paper.

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

## Acknowledgements

We would like to thank Ashraf Al Ashhab and Amatzia Wilk for their help with sample logistics at the Al-Bireh and Shafdan plants, respectively, and Julia Johnke for technical assistance. This project was supported by a grant from the Deutsche Forschungsgemeinschaft (DFG; number CH 731/2-1) and by the Israel Science Foundation grant 1583/12. We thank the Research Center for Agriculture, Environment and Natural Resources at the Hebrew University for funding publication costs.

## Author contributions

Y.C. performed most of the experimental work and analyses, and was the main contributor in writing the manuscript; ZP provided important support in bioinformatics; S.M., TH and FS devised, supported, and participated in the FACS and FC experiments and their analyses and participated to the writing; KKS performed the predation experiments, AAR performed water analyses; AC participated to the design of the work and to the writing, EJ designed and supervised the work and lead the writing of the manuscript.

## Competing interests

The authors declare no competing interests.
