## [Peer Review File · Nature Communications]

Community and single cell analyses reveal complex predatory interactions between bacteria in high diversity systemsReviewers' comments:

Reviewer #1 (Remarks to the Author):

This manuscript analyzes the role of predator-prey interactions in a one-year survey of three wastewater treatment plants (WWTPs) using a high-throughput sequencing approach. The authors validate their sequence-based results with an approach based on FISH tagging and cell sorting, which allowed identification of the interacting predators and preys. In my opinion, the article shows novel and interesting results, while offering significant conclusions in the field of Microbial Ecology, and deserves to be published in Nature Communications. Nevertheless, I have several concerns that probably the authors could discuss and include in the current version of the manuscript.

The PCR approach has potential biases associated to gene amplification such as misrepresentation of the relative abundances of certain populations. In order to validate the findings of the network analysis, an experimental design based on in situ hybridization combined with flow cytometry was developed, giving consistency to the conclusions presented in the manuscript. However, I think that metagenomics, a PCR-independent approach, could also be an option the authors could have used to assess the study of predator-prey interactions and to construct networks, although it is true that this methodology may hamper performing in-depth studies of a particular group, especially from those that are not very abundant. Nevertheless, the authors should at least examine in the discussion section the possibility of comparing metagenomics with amplicon sequencing, since this approach could allow them to explore possible amplification biases as well as to have a better idea of the advantages and limitation of each method.

Also, the authors perform a qPCR analysis to measure absolute abundances of total bacteria, Bdellovibrionales (Bd) and Bacteriovorales (Bx) in one of the WWTPs (Shafdan), while they only measured total bacteria and Bd in Al-Bireh WWTP. Why are there no data for Langenreichenbach WWTP? Besides, I am not able to find the explanation of this methodology in the Materials and Methods section.

Minor comments

P4 L87. Why is Table S2 before Table S1? Please change.

P12, L251-257. This paragraph, together with Fig. 8, describes the different families of preys identified in the FISH-cell sorting experiment. However, in other figures, such as Fig. 4 or 5, microorganisms are described in the order level. This is not the best way to compare between figures. Better use the same phylogenetic level for the sake of comparison, and change the text accordingly. On the other hand, the authors state that 23.1% and 11.1% of the OTUs found in the FISH-cell sorting experiment are common to those found in networks. Why do they find such low values? Can they give an explanation for this result?

Dr. Olga Sánchez

Reviewer #2 (Remarks to the Author):

This manuscript measures the changing species diversity in wastewater treatment plants using a variety of molecular methods.

I am sorry but it was unclear to this reviewer at several points in the manuscript:-

a) Whether novel hypotheses are being tested?

b) Why the narrative at times made microbiologically obvious statements as if they were

unexpected or new ideas?

Examples

Line 27 Does this manuscript show that BALOs control prey populations in a density-dependent manner? If so from which data sets please?

Line 39 What are the foods in the food-web stability in this manuscript? Live bacteria? Dissolved organic carbon? Foods for which organisms? Its unclear.

Line 46 This sentence is problematic- By whom are protists considered generalists and phages mostly specialists? Why "may" they be contributors to bacterial mortality?? What is turnover in this sense? Are bacterial mortal? Can this be rephrased more realistically?

Line 77. Why hypothesize that niche differentiation occurs and is important in a wastewater treatment plant? Any applied microbiology textbook will explain the different microbial populations on the flocs versus in the water (or liquor)itself. This seems like a totally expected thing not a hypothesis.

c) Why the data on prey networks was inferred rather than experimentally tested given that it was important.

This is crucial as many or all of the novel prey species named have no scientific evidence as prey for BALOS and may even lack the correct surface chemistry to be recognised as prey. The manuscript doesnt show they are prey but assumes they are and then writes about discoveries in predator and prey distribution in niches in the wastewater in a scientifically unjustified manner.

Figure 3 could be affected by many many events in the wastewater treatment plant. Cleaning the sewers with disinfectants, drug companies releasing antibiotic waste products into the drains, rainwater run off, fertilizer run off from farming at different times of the year. Why is it showing anything relevant to the experiments?

Line 146 -179 just does not make me think that predator-prey interactions are being identified here.

Why not experimentally test for them?

Please explain- Isn't this just different microbial capabilities to survive fertilizer run off or rainwater dilution or disinfectants or seasonal temperatures?

d) How the experimental results fit with the conclusions drawn or assertions made?

Generally there is not a good description of the limitations and controls of (line 421) "our novel cell sorting approach". It is not clear what percentage of nucleoids could result from wastewater bacteria...and how many were lost inside cell debris...more complex cell architectures as is the case for some of the different bacterial groups they discussed or predatory bdelloplasts trapping nucleoid materials or clay containing particles in the wastewater adsorbing them? So how reliably were data gathered? This is so important to the conclusions but isnt at all explained well.

Line 185 How can this be entitled Predator-prey interactions: prey range- when not a single predation experiment has been done?

The term predators exhibited prey ranges of varied width seems to have no meaning in science? What is the width compared to a bacterium less than a micrometer wide?

Lines 237-238 This seems to make no sense. If there are Firmicutes in the sample (which are not BALO prey) then how does this "provide further support that only predators and attacked prey were present in these sorted samples". It makes no sense at all.

Lines 243-257 Lists of bacteria that are not "standard" prey for BALOs are then generated from

the data. Is this not the time to start testing if these can ever be prey for BALOs? How to write about them being prey when no proof at all is shown in the paper?

e) What the actual outcomes of the work are?

I feel bad that solid conclusions from this work were not evident in the manuscript, as I appreciate what a lot of experimental work was done. There is however a disconnect between the data and the narrative.

Reviewer #3 (Remarks to the Author):

Cohen et al. attempt to resolve predator-prey interactions using co-occurrence analysis (based on Kendall correlations) as well as cell sorting. While the former statistical analysis provides a first hypothesis on potential interactions (these might range from predation to mutualism or might be simple the result of environmental sorting) the later could provide insights into physical association.

The authors intend to identify potential prey and quantify predation by the obligate predators *Bdellovibrio* and like organisms - termed BALOs. However, I am not convinced that the methods used allow to make such inferences. Single cell analysis may allow for the identification of physically associated organisms though this requires validation of the method. Multiple cells can be co-sorted and thus result in presumable co-association. The authors can check for such events using controls or statistical analysis as well as validated association by microscopy. For example the authors could assess if co-isolation of rRNA from different species are not due to chance; that co-isolation of rRNAs from different species by flow cytometry is overrepresented in combination with BALOs, experimental validation with resistant vs. susceptible population, etc. Until results from such additional tests are provided the method can be questioned. Testing total DNA vs. DNA from mda amplified sorted cells is not a valid test - the mda could be explain the difference. What is encouraging and a first strong indication that the method works is that specific sorted populations share a set of associated cells. Even if physical association can be validated, physical association does not imply predatory-prey interactions. An additional test might be to validate bdelloplast formation.

Furthermore, the study is highly focused on WWTP communities and interactions and thus I wonder if it could be of interest to readers outside the field.

Minor comments:

L 26 Please use the term "wastewater treatment plant" instead.

Discussion - statistical association such in the case of correlation analysis cannot be interpreted as predation even when one of the partners belongs to a taxonomic group described to be a BALO. The authors should be aware of the issues of inferring interactions from statistical approaches based on time-series data and inferring ecology such as predation from sequence similarity of a single marker gene. An analysis based on such results does not provide proof - just an hypothesis and in the case of BALOs confirm their potential role as predators and allows the identification of potential prey.

We are indebted to the reviewers for the thorough work they have done which greatly helps improve our manuscript.

Please, find below a point to point answer to each reviewer. The literature cited is numbered with the relevant citations at the end of each reviewer's section.

Reviewer #1 (Remarks to the Author):

This manuscript analyzes the role of predator-prey interactions in a one-year survey of three wastewater treatments plants (WWTPs) using a high-throughput sequencing approach. The authors validate their sequence-based results with an approach based on FISH tagging and cell sorting, which allowed identification of the interacting predators and preys. In my opinion, the article shows novel and interesting results, while offering significant conclusions in the field of Microbial Ecology, and deserves to be published in Nature Communications. Nevertheless, I have several concerns that probably the authors could discuss and include in the current version of the manuscript.

We thank the reviewer for her/his support. We think we have addressed all the concerns mentioned below.

The PCR approach has potential biases associated to gene amplification such as misrepresentation of the relative abundances of certain populations. In order to validate the findings of the network analysis, an experimental design based on in situ hybridization combined with flow cytometry was developed, giving consistency to the conclusions presented in the manuscript. However, I think that metagenomics, a PCR-independent approach, could also be an option the authors could have used to assess the study of predator-prey interactions and to construct networks, although it is true that this methodology may hamper performing in-depth studies of a particular group, especially from those that are not very abundant.

Although 16S rRNA gene community analysis is a very well established, standard approach, the reviewer is right that additional methods can be employed. But the reviewer also mentions the drawbacks of the one he/she proposes. We would like to add that metagenomics (we understand the reviewer meant full metagenomics, i.e. obtaining the gene complement of the bacterial communities) would generate enormous amounts of unnecessary data in the form of genes and pathways not relevant to the questions asked.

Nevertheless, the authors should at least examine in the discussion section the possibility of comparing metagenomics with amplicon sequencing, since this approach could allow them to explore possible amplification biases as well as to have a better idea of the advantages and limitation of each method.

Surely, metagenomics can help identify unknown species but in the case of predators, the function may remain covered. Metagenomics are now mentioned in the discussion: "Recent large scale metagenomics studies reveal up to 5% novel bacterial taxa are missed from 16S rRNA gene community analyses in WW^{30,31}. This may include unknown obligate predators and prey and thus further increase predator and prey diversity. (L270-273).

As for amplification biases, mda/PCR may generate different biases than those caused by the DNA extraction and 16S rRNA gene amplification methods used with the samples upon which the computations (networks) were performed. Yet, both yield similar results, showing that these putative biases did not affect the outcome. We interpret that as further support for the robustness of our approach. Nonetheless, this point is now addressed: "Lastly, FISH-FACS employs different DNA extraction/PCR approaches than the one used to obtain sequences for community analysis. Yet, they support one another." (L405-407).

Also, the authors perform a qPCR analysis to measure absolute abundances of total bacteria, Bdellovibrionales (Bd) and Bacteriovorales (Bx) in one of the WWTPs (Shafdan), while they only measured total bacteria and Bd in Al-Bireh WWTP. Why are there no data for Langenreichenbach WWTP? Besides, I am not able to find the explanation of this methodology in the Materials and Methods section.

A number of samples from Langenreichenbach and Al-Bireh did not yield sufficient DNA to perform all the analyses, so we had to prioritize. A previous study by the EJ team in Zero-Discharge Systems operating with freshwater vs seawater in which fish is grown under high density similarly showed that total BALOs in the different habitats constituting the systems and at different times ranged from 0.1 to 1%³. We thus feel comfortable to generalize that the pattern observed in the Shafdan and Al-Bireh is also relevant for Langenreichenbach. A description of the method has been added to the materials and methods section (L.520-540).

Minor comments

P4 L87. Why is Table S2 before Table S1? Please change.

Corrected.

P12, L251-257. This paragraph, together with Fig. 8, describes the different families of preys identified in the FISH-cell sorting experiment. However, in other figures, such as Fig. 4 or 5, microorganisms are described in the order level. This is not the best way to compare between figures. Better use the same phylogenetic level for the sake of comparison, and change the text accordingly.

Figure 4 has been corrected to present the taxa at the family level. We chose to keep figure 5 at the order level for two reasons: 1. It is not also possible to obtain this information for all OTUs, as there are sequences of poorly defined clades, and; 2. More practically it would make the graphs too dense and difficult to read. Moreover, we think that the order level is very informative by itself as among the orders found in the analyses some have not be described as including BALO preys.

On the other hand, the authors state that 23.1% and 11.1% of the OTUs found in the FISH-cell sorting experiment are common to those found in networks. Why do they find such low values? Can they give an explanation for this result?

We beg to disagree with the reviewer. These values are by no means low for a number of reasons:

1. The samples used for network analysis and in FISH-FACS originated from different WWTPs and were taken at different times: They therefore differ in community composition, and under

such sampling schemes, large differences in OTUs are always observed^{4,5,6}. Yet, prey OTUs that do not overlap between plants mostly belong to the same taxa.

2. Within the same plant, OTUs shift in time within taxa at the family, genus and when the resolution enables the analysis, even at the species level. The functions and structures of the communities while well preserved always differ between plants, and this is largely reflected at the OTU level. It is thus a very supportive result to find about a quarter of conserved prey OTUs between the Eilenburg and Langenreichenbach plants. It can be interpreted as a further confirmation of the biogeographic signature by which closer WWTPs resemble each other more than distant ones, as we also see in this study as well in many others^{4,5,6}.

3. As mentioned in a previous query, the methods can introduce biases: the samples in the network analysis were separated between flocs and liquor. In the FISH-FACS we only used flocs. They were not processed the same way, e.g. deflocculation was performed differently and thus DNA may be differentially extracted. PCR was very different, with mda being used in the FISH-FACS experiment.

Dr. Olga Sánchez

Cited works

1. Eloë-Fadrosh, E. A., Ivanova, N. N., Woyke, T. & Kyrpides, N. C. Metagenomics uncovers gaps in amplicon-based detection of microbial diversity. *Nature Microbiology* 1, 15032, doi:10.1038/nmicrobiol.2015.32 (2016).
2. Parks, D. H. et al. Recovery of nearly 8,000 metagenome-assembled genomes substantially expands the tree of life. *Nature Microbiology* 2, 1533-1542, doi:10.1038/s41564-017-0012-7 (2017).
3. Kandel PP, Pasternak Z, van Rijn J, Nahum O & Jurkevitch E (2014) Abundance, diversity and seasonal dynamics of predatory bacteria in aquaculture zero discharge systems. *FEMS Microbiology Ecology* 89: 149-161.
4. Cohen, Y. et al. Bacteria and microeukaryotes are differentially segregated in sympatric wastewater microhabitats. *Environmental microbiology* 21, 1757-1770 (2019).
5. Saunders, A. M., Albertsen, M., Vollertsen, J. & Nielsen, P. H. The activated sludge ecosystem contains a core community of abundant organisms. *ISME J* 10, 11-20, doi:10.1038/ismej.2015.117 (2016).
6. Wang, X., Hu, M., Xia, Y., Wen, X. & Ding, K. Pyrosequencing Analysis of Bacterial Diversity in 14 Wastewater Treatment Systems in China. *Applied and Environmental Microbiology* 78, 7042-7047, doi:10.1128/aem.01617-12 (2012).

Reviewer #2 (Remarks to the Author):

This manuscript measures the changing species diversity in wastewater treatment plants using a variety of molecular methods.

I am sorry but it was unclear to this reviewer at several points in the manuscript:-

a) Whether novel hypotheses are being tested?

We thank the reviewer for the request. It indeed pushed us to improve this important paragraph.

We rewrote the last paragraph of the introduction and hope this is now clear: "Here, we specifically analyzed the community dynamics of the two major BALO clades (the Bdellovibrionales and the Bacteriovorales) and their association with prey over a year, at three wastewater treatment plants (WWTPs) and invoke ecological theory to explain our results. WWTPs are crucial for keeping public health and reducing environmental pollution¹⁸ and being the most microbe-diverse human-made habitat, they sustain numerous interactions, including predatory interactions^{14,19,20}. We thus ask whether this diversity is also found in BALOs, what the predators' natural prey are and what mechanisms explain their co-existence. We thus hypothesize and show through co-occurrence analysis²¹ that niche differentiation, including prey range partitioning and microhabitat differentiation^{22,23} as well as fluctuating predator and prey populations sustain predator diversity. Finally, in order to validate sequence-based computing results, we developed a direct approach based on FISH tagging²⁴ and cell sorting to identify the interacting predators and prey." (L68-79).

b) Why the narrative at times made microbiologically obvious statements as if they were unexpected or new ideas?

Examples

Line 27 Does this manuscript show that BALOs control prey populations in a density-dependent manner? If so from which data sets please?

*Populations at the taxon level (order, family) cannot be said to be preyed upon in an abundance/density-dependent manner. However, preyed-upon populations at the OTU level are consumed in a density-dependent, KtW-like manner. This is because different prey strains can be largely different in terms of susceptibility to predation as shown by^{8,9} and in our hands at well. To illustrate this point, within the species *Acidovorax citrulli* for example, different strains are "immune" to the BALO predators while others are susceptible. Please see the figure below with two resistant and two sensitive strains of *A. citrulli*, tested against three different predators (this is taken from another ongoing research project).*

We understand that there might have been a discrepancy between the abstract and a sentence at lines 195-196 (first submission line numbers) and both have been slightly amended to make the point clearer. This includes, in the main text: "Lower abundance taxa (<3%) like Myxococcales and Rhizobiales (Figure S6) could contribute a rather important fraction of the predator-prey edges (5.8±3.8% and 3.1±2.4%, respectively), while clades like Rhodocyclales appeared to be proportionally less preyed upon than their relative abundance would tell (4.9±2.33% predation for a relative abundance of 11.7±3.8%). Yet, at the prey OTU level, correlation analysis shows BALO predation to be density-dependent (Figure 4)." (L176-181).

Caption: *A. citrulli* strains 7a1 and W1 are preyed upon by *B. bacteriovorus* strains HD100 (HD), 109J (109), and BER2, while strains M6 and M1 are resistant.

Line 39 What are the foods in the food-web stability in this manuscript? Live bacteria? Dissolved organic carbon? Foods for which organisms? Its unclear.

This was an introductory sentence presenting a large ecological question. Yet, to dismiss any misunderstanding it now states "trophic web".

Line 46 This sentence is problematic- By whom are protists considered generalists and phages mostly specialists? Why "may" they be contributors to bacterial mortality?? What is turnover in this sense? Are bacterial mortal? Can this be rephrased more realistically?

There is ample literature that relates to protists and phages as generalists and specialists, respectively. These definitions, like all definitions are broad-brush and of course, any protist does not prey upon all bacteria and any phage does not infect only one particular lineage of bacteria¹⁰⁻¹². Yet, predatory protists as a general rule, have much larger prey ranges than phages. We used this accepted approach to introduce the BALOs which, based upon our current knowledge show intermediate prey ranges in terms of the diversity of prey they can use. One of the findings of our study is that indeed, using a culture-independent approach, we can show that BALO prey ranges largely vary. Yes, bacteria are mortal, they can die.

Line 77. Why hypothesize that niche differentiation occurs and is important in a wastewater treatment plant? Any applied microbiology textbook will explain the different microbial populations on the flocs versus in the water (or liquor) itself. This seems like a totally expected thing not a hypothesis.

What is expected and appears to be obvious is often a very complex matter, and as long as it is now demonstrated it is a hypothesis at best. As far as we know, no research has addressed

the question of co-existence of these micro-predators. Actually, no research has shown there ARE so many BALOs in any habitat. The question then being how they co-exist if they all prey on gram negative bacteria becomes an important one. Niche differentiation is a basic ecological mechanism that can explain co-existence of organisms performing identical functions in the same environment. Without some kind of difference, e.g. prey range, temporal functioning, or spatial separation, less competitive species are out-competed by the fittest one. As for sympatric differentiation between flocs and liquor microorganisms in WWTPs, this is the main topic of two recent papers from our groups^{7,13} that are amply cited in the manuscript. The Cohen et al (2019) article shows in detail the differences and the overlaps between different types of microorganisms in flocs and liquor and we show that while the two microhabitats share similar taxa, the distributions of very specific strains within species are different. Furthermore, the data shown in the present manuscript clearly demonstrates that BALOs behave otherwise and do not differentiate between the microhabitats. To put in an applied perspective, in the type of plants we worked with, flocs are the active part of the water purification process while the liquor becomes the effluent. We are sure that the reviewer agrees with us that the microbiology of flocs and liquor is not so obvious and that not everything is already written in the textbooks (which by the way have to be re-written from time to time).

c) Why the data on prey networks was inferred rather than experimentally tested given that it was important.

We do not understand the comment. The prey network was indeed experimentally tested by the FISH-FACS experiment. This latter experiment used a totally different methodology and was carried out independently to the network analysis. More so, it made use of another set of samples.

This is crucial as many or all of the novel prey species named have no scientific evidence as prey for BALOS and may even lack the correct surface chemistry to be recognised as prey. The manuscript doesn't show they are prey but assumes they are and then writes about discoveries in predator and prey distribution in niches in the wastewater in a scientifically unjustified manner.

We do not know of studies showing what the surface chemistry of BALO prey is. Actually, this is a Holy Grail of the field: how BALOs recognize their prey. So far, no one knows.

More specifically, the use of co-occurrence networks in ecology, including microbial ecology to infer interactions between species, is not only an accepted approach, it is a very useful one to generate further working hypotheses for field surveys and experiments. However, we would certainly agree with the reviewer that computational inference is not a proof. That was the reason why we developed the FISH-FACS approach.

Although it is an impossible task to isolate the potential prey and their predators we understand the reviewer's concern. We thus used model BALOs from the lab on bacterial species isolated from WWTPs to test if the latter could indeed be preyed upon by BALOs. Please see the full answer in a related query below (query " Line 146 -179 just does..."). The revised version of the manuscript includes the results of these experiments.

Figure 3 could be affected by many many events in the wastewater treatment plant. Cleaning the sewers with disinfectants, drug companies releasing antibiotic waste products into the

drains, rainwater run off, fertilizer run off from farming at different times of the year. Why is it showing anything relevant to the experiments?

This is why environmental parameters are measured along with the microbiome, so fluctuations in time could be related, correlated to extrinsic abiotic parameters (e.g. temperature, pH, EC, etc.) or to biotic parameters (e.g. prey abundance and diversity). But the reviewer is right that a part of the variance remains unknown. That is a problem not only in this analysis but in the analysis of any ecosystem, and the reason why researchers often use microcosms or other very well controlled settings in which sewers and industry cannot affect the results.

To be able to find the sources of variance, we defined a large set of measurements and show that among the abiotic parameters only temperature has a significant, overall effect. Other parameters may explain part of the variance but not at the same scale and not across the experiment. That is why we do not address them.

Moreover, disinfectants, antibiotics or other chemicals entering the system in high quantities would not be expected to be so selective as to affect specific strains of BALOs but no other strains, and also no other microbe, as we did not observe such fluctuations in other groups⁷. If that were the case, we can reasonably postulate that the WWTP operation itself would suffer and that would be readily detected in environmental parameters like COD, BOD, nitrogen concentration etc. But certainly we cannot reject all unfalsifiable hypotheses.

Line 146 -179 just does not make me think that predator-prey interactions are being identified here.

Why not experimentally test for them?

*If the reviewer means to isolate the potential prey and their predators that would be an impossible task. We understand the reviewer's concern to have at least a proof of concept that "novel" prey are indeed true prey of BALOs. We did go along this idea and used model predators we have in the lab to test their ability to utilize prey strains isolated from WWTPs that have hitherto not been shown to be preyed upon by BALOs. For that purpose, we used different predators, including two strains expressing the Tdtomato fluorescent protein to track their growth in the presence of the tested strains. This revealed that both *B. bacteriovorus* HD100 and 109J expressing this protein grow on *Zooglea*, *Azoarcus* and *Thaurea* strains but not on *Acidovorax caeni*, *Haliscomenobacter hydrossis* or on the other *Thaurea* strains tested. Further, a *Peridibacter* strain (*Bacteriovoraceae*) preyed upon *Z. caeni* and an *Azoarcus* strain, and *B. exovorus* JSS, an epibiotic predator isolated from WW so far known to only prey on two bacterial strains, preyed upon *Z. caeni* (Figure S7 in the resubmitted manuscript). We thus show that BALOs are indeed able to prey upon strains belonging to clades present in WWTPs that had not been known to be preyed upon based on previous culture-based experiments. We would like to add that other studies showed, by spiking BALOs into WW samples and examining community structure by 16S rRNA gene sequencing, that similar taxa as detected in our study were significantly affected by the introduced predators^{14,15}. As the computational and sorting-based analyses are backed by bench experiments, we are completely confident that the "novel prey" are indeed true prey for BALOs. Please see lines 189-193.*

Please explain- Isn't this just different microbial capabilities to survive fertilizer run off or rainwater dilution or disinfectants or seasonal temperatures?

We think that this question is very similar to one asked – and answered above- concerning the drivers behind the observed dynamics.

d) How the experimental results fit with the conclusions drawn or assertions made?

Generally there is not a good description of the limitations and controls of (line 421) "our novel cell sorting approach". It is not clear what percentage of nucleoids could result from wastewater bacteria...and how many were lost inside cell debris...more complex cell architectures as is the case for some of the different bacterial groups they discussed or predatory bdelloplasts trapping nucleoid materials or clay containing particles in the wastewater adsorbing them? So how reliably were data gathered? This is so important to the conclusions but isn't at all explained well.

In this study flow cytometry clearly separated the BALOs by light scatter behavior and the fluorescent label out of the cell debris and instrumental noise. Particles frequently have an overall distribution according to scatter and auto-fluorescent characteristics in the histograms, they do not cluster and are separated from the cells by the cell handling procedure. Therefore, particles were not part of the sorting gate. The negative control is shown in Figure S9B.

In addition, we want to point out that cell sorting increases the information and the resolution on present taxa in a gate because the further handling of sorted cells (i.e. 16S analysis) excludes background information caused by the whole (unsorted) sample¹⁶.

The reviewer is right that not all DNA which is extracted from any sorted sample may be detectable because some cells are more difficult to pry open than others. It can take many rounds of DNA extraction to arrive to saturation¹⁷. That is certainly true for our particular study as it is for the vast majority of microbiome studies. This is one of the limitations of nucleic acids-based methods and a sentence has been added to convey this limitation: "Our novel cell sorting approach although still limited by methodological constraints like DNA extractability⁷⁶ and the number of retrieved cells⁸ and the number of retrieved cells, fills a crucial knowledge gap and provides the mechanistic details needed to quantify BALO-imposed bacterial mortality, and their contribution to microbial turnover in trophic networks" (L415-418).

Line 185 How can this be entitled Predator-prey interactions: prey range- when not a single predation experiment has been done?

We disagree. The whole study shows not one but numerous predator-prey interactions. We infer from the comment that the reviewer regrets that no bench experiment with isolated predator and prey were performed. Accordingly, this is now included, as described above. Yet, the study's aim is to overcome the very constraining limitations of culture-dependent experiments of BALO-prey interactions, and to reveal their dynamics in situ in a widely distributed, complex ecological system. BALOs are difficult to isolate, requiring rounds of purification, and often the use of enrichment. In most studies based on the isolation of BALOs, only a few strains are isolated. Furthermore, as a defined prey has to be used, BALOs not preying on the proposed strain are not seen. And as most bacteria are not culturable, the true diversity of BALOs in any sample is only for one to guess. Our study overcomes these limitations. We certainly agree that our task is to show that our approach is valid, and enables to precisely disentangle predator-prey interactions from the mass of data. We are convinced

that we indeed show this as detailed in the manuscript and in answers to the reviewers' questions.

We welcome the reviewer' queries for clarifications, as they improve our manuscript. However, one would understand we cannot accept the statement that no predator-prey experiment has been done, de facto negating all this work in less than twenty words.

The term predators exhibited prey ranges of varied width seems to have no meaning in science? What is the width compared to a bacterium less than a micrometer wide?

The term is indeed confusing, we now use "wide" and "narrow" prey ranges.

Lines 237-238 This seems to make no sense. If there are Firmicutes in the sample (which are not BALO prey) then how does this "provide further support that only predators and attacked prey were present in these sorted samples". It makes no sense at all.

We thank the reviewer for the comment that helps clarify the text. Our intent was to show that G+ cells are extremely rare in the BALO-sorted samples. The sentence has been rewritten to clarify this, and it now is:

"High throughput 16S rRNA gene sequencing of the P1, P14 and P16 samples using universal 16S rRNA-gene primers revealed depletion for gram positive taxa in samples P1 and P14 at relative read abundances of $<6.10^{-6}$ and $<3.10^{-4}$, respectively, compared to about 6.10^{-2} in the P16 sample, providing further support that predators and attacked prey (gram negative) were very largely enriched in the BALO-sorted samples (Figure 8A, $p<0.01$)." (L223-227).

Lines 243-257 Lists of bacteria that are not "standard" prey for BALOs are then generated from the data. Is this not the time to start testing if these can ever be prey for BALOS? How to write about them being prey when no proof at all is shown in the paper?

Sure, and again thank you for suggesting this. It has been added, as described above.

e) What the actual outcomes of the work are?

I feel bad that solid conclusions from this work were not evident in the manuscript, as I appreciate what a lot of experimental work was done. There is however a disconnect between the data and the narrative.

We are puzzled by this statement. The approach described in our work is rather "standard" in the sense that 16S rRNA gene community analysis enables, among other things, addressing the unculturable bacteria, identify them, track them, and infer their function. With these data at hand, one can understand the dynamics of communities, and how biotic and abiotic parameters interact with its members to shape responses/functional outputs. We applied it to BALOs in wastewater. Unless one does not trust the technologies and resulting analyses, we think that the conclusions from our work are novel, solid and far-reaching. We reiterate a point made earlier: it is now possible to analyse microbial predator-prey interactions in complex ecosystems in a culture-independent way. The huge BALO diversity revealed here shows how far we are from understanding what BALOs can do in nature only using cultures.

Cited papers

1. Daims, H., Taylor, M. W. & Wagner, M. Wastewater treatment: a model system for microbial ecology. *Trends in Biotechnology* 24, 483-489, (2006).
2. Feng S, Tan CH, Constancias F, Kohli GS, Cohen Y & Rice SA (2017) Predation by *Bdellovibrio bacteriovorus* significantly reduces viability and alters the microbial community composition of activated sludge flocs and granules. *FEMS Microbiology Ecology* 93: fix020-fix020.
3. Dolinšek, J., Lagkouvardos, I., Wanek, W., Wagner, M. & Daims, H. Interactions of Nitrifying Bacteria and Heterotrophs: Identification of a *Micavibrio*-Like Putative Predator of *Nitrospira* spp. *Applied and Environmental Microbiology* 79, 2027-2037, doi:10.1128/aem.03408-12 (2013).
4. Yu, R., Zhang, S., Chen, Z. & Li, C. Isolation and application of predatory Bdellovibrio-and-like organisms for municipal waste sludge biolysis and dewaterability enhancement. *Frontiers of Environmental Science & Engineering* 11, 10, doi:10.1007/s11783-017-0900-3 (2017).
5. Welsh, R. M. et al. Bacterial predation in a marine host-associated microbiome. *ISME J*, doi:10.1038/ismej.2015.219 (2015).
6. Pineiro, S. et al. Niche Partition of Bacteriovorax Operational Taxonomic Units Along Salinity and Temporal Gradients in the Chesapeake Bay Reveals Distinct Estuarine Strains. *Microbial Ecology* 65, 652-660, doi:10.1007/s00248-013-0186-3 (2013).
7. Cohen, Y. et al. Bacteria and microeukaryotes are differentially segregated in sympatric wastewater microhabitats. *Environmental microbiology* 21, 1757-1770 (2019).
8. Dashiff A, Junka R, Libera M & Kadouri D (2011) Predation of human pathogens by the predatory bacteria *Micavibrio aeruginosavorus* and *Bdellovibrio bacteriovorus*. *J Appl Microbiol* 110.
9. Shanks, R. M. Q. et al. An Eye to a Kill: Using Predatory Bacteria to Control Gram-Negative Pathogens Associated with Ocular Infections. *PLoS ONE* 8, e66723, doi:10.1371/journal.pone.0066723 (2013).
10. Edwards, R. A., McNair, K., Faust, K., Raes, J. & Dutilh, B. E. Computational approaches to predict bacteriophage–host relationships. *FEMS microbiology reviews* 40, 258-272 (2016).
11. Fenchel, T. *Ecology of Protozoa: The biology of free-living phagotropic protists*. (Springer-Verlag, 2013).
12. Johnke, J. et al. A generalist protist predator enables coexistence in multitrophic predator-prey systems containing a phage and the bacterial predator bdellovibrio. *Frontiers in Ecology and Evolution* 5, 124 (2017).
13. Šimek, K. & Chrzanowski, T. H. Direct and indirect evidence of size-selective grazing on pelagic bacteria by freshwater nanoflagellates. *Applied and Environmental Microbiology* 58, 3715-3720 (1992).
14. Feng S, Tan CH, Cohen Y & Rice SA (2016) Isolation of *Bdellovibrio bacteriovorus* from a tropical wastewater treatment plant and predation of mixed species biofilms assembled by the native community members. *Environmental microbiology* 18: 3923-3931.
15. Feng S, Tan CH, Constancias F, Kohli GS, Cohen Y & Rice SA (2017) Predation by *Bdellovibrio bacteriovorus* significantly reduces viability and alters the microbial community composition of activated sludge flocs and granules. *FEMS Microbiology Ecology* 93: fix020-fix020.
16. Cichocki, N. et al. Bacterial mock communities as standards for reproducible cytometric microbiome analysis. *Nature Protocols* 15, 2788-2812, (2020).
17. Delmont TO, Robe P, Cecillon S, Clark IM, Constancias F, Simonet P, Hirsch PR & Vogel TM (2011) Accessing the Soil Metagenome for Studies of Microbial Diversity. *Applied and Environmental Microbiology* 77: 1315-1324.

Reviewer #3 (Remarks to the Author):

Cohen et al. attempt to resolve predator-prey interactions using co-occurrence analysis (based on Kendal correlations) as well as cell sorting. While the former statistical analysis provides a first hypothesis on potential interactions (these might range from predation to mutualism or might be simple the result of environmental sorting) the later could provide insights into physical association.

The authors intend to identify potential prey and quantify predation by the obligate predators *Bdellovibrio* and like organisms - termed BALOs. However, I am not convinced that the methods used allow to make such inferences. Single cell analysis may allow for the identification of physically associated organisms though this requires validation of the method. Multiple cells can be co-sorted and thus result in presumable co-association.

Flow cytometry is a method that measures single cells. Doublets can only be sorted together if they are strongly and physically connected and small enough to pass a 10 µm sample stream. Doublets would be easily detectable by their position in the histogram afar from the individuals. Larger aggregates of cells, if they are small enough to be measurable by flow cytometry, do not form virtual cells clusters that can be gated. They would all be of a different size and fluorescent intensity and distribute randomly in the upper right corner of the histogram, something that we did not see. Therefore, microbial cytometry goes for single cells in an identical way as medical flow cytometry goes for single cells. We do not co-sort multiple cells. This is also plainly seen by the pulse – width plot, which is usually used in eukaryotic cell ploidy and aggregate approaches to separate those from individuals. Our pulse-width plot does not show any multiple-cell events. Please see the microbial community cytometry methodology explained in a recent publication in Nature Protocols by the SM team ¹.

The authors can check for such events using controls or statistical analysis as well as validated association by microscopy. For example the authors could assess if co-isolation of rRNA from different species are not due to chance; that co-isolation of rRNAs from different species by flow cytometry is overrepresented in combination with BALOs, experimental validation with resistant vs. susceptible population, etc.

We thank the reviewer for the comment that led us to add controls (see below) and thus, improve our manuscript.

Our results are well supported statistically, a statement further supported by the MRPP-A analysis on the Bray-Curtis similarities between samples that has been added to figure 8. It shows high dissimilarity between BALO-sorted samples and the other samples with very significant p values.

Cell sorting is a subsampling method, differentiating the majority of cells as background from the sorted cells. If gates are set for the whole community and 16S analyses performed, the sum of sorted cells from the various gates will always show a higher resolution in OTUs compared to the whole community. Cell sorting is a physical approach, cells separated in this way from the major community are separated with purity mode of 99%. This purity mode fits with the threshold that is used for the evaluation of the 16S data, therefore, this 1 % possible failure by cell sorting does not interfere with the usual interpretation of 16S data.

The co-sorting of BALOs with prey via respective gate setting did indeed show an increased amount of BALOs per gate in comparison to their representation in the whole community (15-24% vs. 0.1-1%). By using the '1.0 Drop Pure' sort mode the failure of missorting is around 1%.

500,000 cells are sorted for the successive 16S analyses. This routine application in our lab is published in numerous papers e.g.^{1,2}.

The gate from which a co-isolation of rRNA from different species along with that from the predator was performed was set for individuals that exhibiting FISH-fluorescence (predator, bdelloplasts, prey-predator attached). This enabled us to find out with which prey the predator associates. Fluorescently labelled cells and cells to which the predator attached or invaded were sorted, and the species in the BALO-associated fractions were very strongly biased towards gram negative cells explaining an underrepresentation of gram positive cells in the positive gate as compared to the controls of the total community and of the sorted negative gate. This latter gate did not contain fluorescent signals, no BALOs were detected in it by sequencing and its community composition was significantly different from that of the samples containing BALOs. We logically infer that the sorted cells are almost all associated with BALOs. These findings are now complemented by two kinds of experiments:

- We tested the growth of two strains of *B. bacteriovorus* expressing a fluorescent reporter gene on a number of strains originating from WW: Three species of *Thauera*, an *Azoarcus*, *Acidovorax caeni*, *Zooglea caeni*, *Haliscomenobacter hydrossis* belonging to taxa hitherto not tested for BALO predation, and *E. coli*. *E. coli*, *Azoarcus olearius*, *T. aromatica*, and *Z. caeni* were preyed upon. Predation upon the *Azoarcus* and the *Zooglea* strains was largely delayed in both predators compared to *E. coli*, and on *T. aromatica* by strain HD100. *A. caeni*, *H. hydrossis* and the two other *Thauera* were not preyed upon by these BALO strains. We also tested a *Peridibacter* strain (*Bacteriovoraceae*) and *B. exovorus* JSS, an epibiotic predator isolated from WW so far known to only prey on two bacterial strains. Both preyed upon *Zooglea*, and *Peridibacter* also preyed on *Azoarcus*. Thus prey strains differ in their ability to sustain the growth of the predators some being "efficient", sustaining rapid growth of the predator or being less susceptible to predation. Hence, the tested predators, as expected have a "versatile" prey range, preying on a number but not all potential prey, with varying efficacies. This is now shown in figure S7 in the revised manuscript. Please see lines 189-193.

-We then tested some of these same prey and non-prey for attachment. Flow cytometry data show that the predator attaches to prey whether it is an "efficient"(*E. coli*) or a less susceptible (*Azoarcus*) prey but not to the non-prey (*Acidovorax*)(Figure S11 in the revised manuscript), mentioned as "A further control for spurious attachment of BALOs to non-prey vs prey strains isolated from WWTPs by BALOs confirmed specificity (Figure S11)". (L216-217).

- We also performed a microscopy-based analysis in which interaction time of the fluorescent predator *B. bacteriovorus* 109J-Tdtomato was measured on prey (*E. coli*, *K. oxytoca*, *Pectobacterium*) and non-prey (*Acidovorax citrulli* M6) showing that the predator did not "stay" on the non-prey cells (see figure below). Why prey are "efficient" or "less susceptible" we do not know. We can postulate that it is due to the kind of "receptors" the predator latches on the prey cell wall or to differential distribution of phenotypically resistant cells³ but this is clearly beyond the scope of this manuscript. Interestingly, bdelloplasts, which are invaded prey and immune to further invasion retained the predator for long. However, in the context of our FISH-FACS experiments this is not an issue. We do not include these data in the manuscript itself as this experiment was performed within another study using different bacterial species. We use it here to support our claim that prey and non-prey interactions are indeed specific.

Caption: One minute-long videos were recorded at 30 frames per second and the predator's swimming paths manually tracked. Handling time is defined as the time spent by the predator on the surface of the cell until detachment. In prey strains (*E. coli*, *K. oxytoca* and *P. carotovorum*) only unsuccessful interaction events are shown, as productive interactions lead to penetration of the prey by the AP cell.

Until results from such additional tests are provided the method can be questioned. Testing total DNA vs. DNA from mda amplified sorted cells is not a valid test - the mda could be explain the difference.

The reviewer is right that mda can cause amplification biases. However, for this to explain our results the bias should be extraordinary as to not only affect BALO sequences (present in one type of samples but not the other) but also many more species. And maybe most important, the bias of BALO amplification would have to be correlated to the intensity of the fluorescent probe used to sort the cells. We think this very highly improbable.

What is encouraging and a first strong indication that the method works is that specific sorted populations share a set of associated cells.

Thank you.

Even if physical association can be validated, physical association does not imply predatory-prey interactions. An additional test might be to validate bdelloplast formation.

We think that the experiments above show beyond any reasonable doubt that BALOs greatly associate more with prey than with non-prey. However we would like to provide the reviewer with a detailed answer why the request would be almost impossible to answer: we understand that the reviewer requires visual proofs beyond FACS in order to show that physical interactions are predatory interactions and not spurious interactions with non-prey. For such results to be valid we would have to show that all (or almost all) BALO-associated cells are prey, with strong statistics. Here is why:

-Detecting bdelloplasts for the sake of it is meaningless: BALOs thrive in the environment, and they are obligate predators which most often penetrate their prey. An anecdotal picture of an "environmental bdelloplast" would thus show the obvious, that BALOs prey to survive in nature. Technically, it would be very difficult to obtain as there is no knowledge on the shape and size of environmental bdelloplasts which may be quite diverse, as even within cultured prey strains, bdelloplasts may have different shapes⁴.

- More importantly, fluorescence sorts AP cells as well as bdelloplasts. In nature, BALO populations are heterogeneous i.e. BALOs are not growing synchronously, they are found in a number of states: as free-swimming AP cells, as AP cells attached to other cells, and within bdelloplasts. If AP cells associated to bacterial cells form more than a few % of the total BALO population, it is impossible to prove the point because we would not be able to tell if any particular cell a BALO is attached to is a prey or not. Thus only bdelloplasts can be proof of selectivity as put forward by the reviewer. But since they only form that much of the BALO population at any time, even if they can be separated and observed would not prove that there is no attachment to non-prey and thus that all of BALO-associated sorted populations are indeed prey to BALOs.

As an example of local dynamics (predation of a specific BALO on a specific prey population), please see the graphs below. They depict a flow cytometric analysis of a synchronous culture of a single predator and a single prey, showing the formation of bdelloplasts and the release of progeny AP cells. In an environmental sample, they would be completely mixed.

To summarize: we show single cell sorting of BALO-associated populations. These data are confirmed by pulse width analyses; by attachment to prey but not to non-prey; by flow cytometry and microscopy, and most importantly, BALO-associated populations are the same whether independently deciphered by FISH-FACS or by correlation (network) analyses. Finally, we show that computed taxa of "novel" prey are preyed upon by BALOs, and point out that they also correlate with BALO occurrence in other studies of WWTPs^{5,6,7}.

Furthermore, the study is highly focused on WWTP communities and interactions and thus I wonder if it could be of interest to readers outside the field.

First, the method is generic. It can be applied to any habitat where BALOs are being studied such as the ocean, soil, guts, fish ponds etc. It can serve as a model not only to track BALO-prey interactions but also for other types of physical interactions between bacteria like parasitism, syntrophy, and else.

Second, wastewater treatment is not a curiosity. It is, along with vaccination and antibiotics the main reason for the dramatic improvement in human health in the last 150 years. WWT is an expensive feature. Reducing costs is thus important, more so to enable Low and Medium Income Countries to gain large-scale access to WWT. This is still a major developmental goal: "Ensure access to water and sanitation for all" is one of the UN sustainable development goals. "Worldwide, one in three people do not have access to safe drinking water, two out of five

people do not have a basic hand-washing facility with soap and water, and more than 673 million people still practice open defecation." <https://www.un.org/sustainabledevelopment/water-and-sanitation/>. This study won't solve this huge problem of course but we hope it can be a small step in the right direction.

Minor comments:

L 26 Please use the term "wastewater treatment plant" instead.
Corrected.

Discussion - statistical association such in the case of correlation analysis cannot be interpreted as predation even when one of the partners belongs to a taxonomic group described to be a BALO.

The authors should be aware of the issues of inferring interactions from statistical approaches based on time-series data and inferring ecology such as predation from sequence similarity of a single marker gene. An analysis based on such results does not provide proof - just an hypothesis and in the case of BALOs confirm their potential role as predators and allows the identification of potential prey.

We indeed are aware that correlations are not proof. Yet the dynamics behavior of the BALO populations (as OTUs) are best explained by their relationship to specific prey OTUs. Other environmental parameters do not explain them except temperature that shows a clear signal upon specific OTUs. As we presented above, at this stage, a definite proof of prey consumption and predator growth at the specific OTU level is not to be achieved. [In an allegorical manner, we have "collared" a large group of predators in a highly populated, dense habitat, tracked them, shown with remote technology that they interact with other species and that the populations of both fluctuate in correlation one to the other. We also know from cage experiments that representative of the predatory group can only survive and multiply while eating prey. But we have not been able to photograph the interactions in the habitat itself.]

Cited papers

1. Cichocki, N. et al. Bacterial mock communities as standards for reproducible cytometric microbiome analysis. *Nature Protocols* 15, 2788-2812, (2020).
2. Liu Z, Cichocki N, Hübschmann T, Süring C, Ofițeru ID, Sloan WT, Grimm V, Müller S. Neutral mechanisms and niche differentiation in steady-state insular microbial communities revealed by single cell analysis. *Environmental Microbiology* (2019), 21/1, 164-181.
3. Shemesh Y & Jurkevitch E (2004) Plastic phenotypic resistance to predation by *Bdellovibrio* and like organisms in bacterial prey. *Environmental Microbiology* 6: 8-12.
4. Tudor, J., McCann, M. & Acrich, I. A new model for the penetration of prey cells by bdellovibrios. *Journal of Bacteriology* 172, 2421-2426 (1990).
5. Feng S, Tan CH, Cohen Y & Rice SA (2016) Isolation of *Bdellovibrio bacteriovorus* from a tropical wastewater treatment plant and predation of mixed species biofilms assembled by the native community members. *Environmental microbiology* 18: 3923-3931.
6. Feng S, Tan CH, Constancias F, Kohli GS, Cohen Y & Rice SA (2017) Predation by *Bdellovibrio bacteriovorus* significantly reduces viability and alters the microbial community composition of activated sludge flocs and granules. *FEMS Microbiology Ecology* 93: fix020-fix020.
7. Jurkevitch, E. in *The Ecology of Predation at the Microscale*. p. 37-64 (Springer, 2020).

REVIEWER COMMENTS

Reviewer #3 (Remarks to the Author):

The authors have done a great job in addressing the methodological concerns raised. Still, the interpretation of the results needs further work.

Throughout the manuscript the authors should avoid using the term predator-prey networks and instead use co-occurrence networks when presenting negative correlations of OTUs in the 16S rRNA gene data. The co-occurrence networks only provide potential predatory-prey interactions. Some of them could be validated by the cell sorting and sequencing approach. It would be good to show some statistics on which ones could be validated - there are some numbers mentioned but it would be of interest to see how this looks for different taxonomic groups, sites (WWTPs) and over the year. I would also suggest to focus on those when interpreting and discussing predatory-prey interactions, such as oscillations.

The density/frequency dependent fluctuations/oscillations of prey and predator as emphasized in the abstract are not well presented in the manuscript - oscillations are mentioned in connection with the network analysis but hard to connect to density dependent fluctuations - as these co-occurrence patterns could be due to environmental selection or simply driven by spurious correlations. How the physical and chemical conditions may result in observed co-occurrence patterns needs to be considered and potential biases discussed in much more detail. As pointed out by one of the reviewers earlier there is no data and analysis presented that would allow to make inferences on density dependent predatory-prey interactions. The results might be hidden somewhere but in the current manuscript this is not conveyed. Maybe focusing on the co-occurrence patterns where you have indications on predatory-prey interactions from the cell sorting results might provide relevant starting points.

Here some more comments:

L 69 You may use "potential prey"

L 70 What ecological theory are the authors inferring here. They should be more specific.

L 74 What results allow to make inferences on the co-existence of diverse OTUs of BALOs?

L 76 I am missing results on the microhabitat differentiation. The authors do not provide any results on this - The analyses performed and presented do not allow to make a very strong point as they do not allow to resolve differential distribution across microhabitats.

L 164-187 The authors have no proof for predation - they have negative correlations that may provide potential prey-predator interactions. This needs to be clarification! The authors can only talk about potential predator-prey interactions.

L 250. How do the authors show dependence upon prey density on shorter time frames? And how is co-existence of numerous predators explained by prey range partitioning? The authors have some indications from the networks based on temporal dynamics but nothing more.

L 288. Here only two habitats were differentiated. Could there be others?

L 305. What were the absolute numbers? Do the authors have some estimates?

L 436 Please state that this is the analysis used for bacterial community analysis in the case of 2 WWTPs.

Figure 3. If these are relative abundances based on read numbers this should be stated in the figure description. And units should be given in the figures.

Figure 6. Is there a way how this can be combined with showing the overlaps in potential prey among the different treatments. Please add how these results were obtained.

Reviewer #4 (Remarks to the Author):

This manuscript describes an investigation of the population dynamics and interactions of the predatory bacteria, *Bdellovibrio* and like organisms (BALOs), specifically *Bdellovibrionales* (Bd) and *Bacteriovoracales* (Bx), in waste water treatment plants. The authors report that the results of their study show greater diversity of the predators than previously known; the population dynamics and interactions of the BALOs with other bacteria in wastewater treatment facilities, the impact of seasonal changes and prey abundance and existence of numerous predators by prey range partitioning. The manuscript has been previously reviewed and the reviewers provided extensive comments. The current review focuses on the authors' responses to the original reviewers' comments and other aspects of the manuscript. The manuscript describes a rather complex study that utilizes many state-of-the-art technologies to answer some critical questions on the interaction of BALOs with other bacteria. The use of flow cytometry to enumerate and determine BALO association with individual cells is a novel approach for the study of the predators. Network analysis was also utilized. These technologies are not without limitations in deciphering BALO interactions, but do provide strong, but presumptive, information based on the experimental data. Nevertheless, the data and the conclusions based on them represent an important advance in the study of BALOs.

The authors' responses to the original reviewers' comments were complete and thorough. The authors did take those comments seriously and in many cases made revisions to the manuscript, and in some cases included results from additional experiments. Although in their responses to the original reviewers' comments, the authors did acknowledge some limitations in some of the methodologies, it would be helpful if they acknowledged more the presumptive nature of their findings and discuss future studies and methodologies to confirm results where appropriate. These changes would be minor involving revisions of a few phrases or words. Several examples are below. The first is on line 250: Suggest that "uncovers" be changed to suggest or indicate.

Line 250: uncovers differential temporal dependence of BALOs upon seasons on the long term and upon
and on line 251 add "appears to" as shown below.

Line 251: prey density on shorter time frames, and; appears to explain co-existence of numerous predators by prey range partitioning.

Another example is shown in line 256.

It's suggested that "precise" be deleted.

Finally, the study establishes a

Line 256: novel approach that enables the precise tracking and identification of interacting predators

A few other comments follow.

In line 437- Green et al., 2015 is missing from reference list. Please check all the references.

Why for some sites they used only BALO specific 16S rRNA gene primer but for the some they used 16S rRNA for total bacteria and BALO both?

The authors should provide the version of the SILVA

Rehovot, 25.03.2021

We feel indebted to the reviewers for their thorough and diligent work that truly contributed to improving the manuscript. We also thank you for the rapid and supportive handling of our submissions. Please, find our point-to-point answers below.

Reviewer #3 (Remarks to the Author):

The authors have done a great job in addressing the methodological concerns raised. Still, the interpretation of the results needs further work.

Thank you.

Throughout the manuscript the authors should avoid using the term predator-prey networks and instead use co-occurrence networks when presenting negative correlations of OTUs in the 16S rRNA gene data. The co-occurrence networks only provide potential predatory-prey interactions. Some of them could be validated by the cell sorting and sequencing approach. It would be good to show some statistics on which ones could be validated - there are some numbers mentioned but it would be of interest to see how this looks for different taxonomic groups, sites (WWTPs) and over the year. I would also suggest to focus on those when interpreting and discussing predatory-prey interactions, such as oscillations.

The term "predator-prey networks" has been replaced by "co-occurrence networks" or "potential predator-prey networks". Within them, specific interactions between predators and defined taxa are already described in detail, including the Myxococcales, Sphingobacteriales, Rhizobiales and Rhodocyclales for which relative abundances vs appearance in potential predatory interactions are mentioned in the text. The statistics, unless we do not understand the reviewer's request are "built in" in the co-occurrence network analysis as they are based on $p < 0.05$ and a Kendal correlation of > 0.7 . Furthermore, a comparison between relative abundance of a clade and potential predatory interactions of this clade with BALOs are also validated statistically (Figure S6).

The reviewer writes " Some of them (i.e. co-occurrence networks) could be validated by the cell sorting and sequencing approach".

Providing full FISH-FACS -based network analysis is much beyond the scope of this research as it would require sorting hundreds of samples. We infer from previous comments that the reviewer is knowledgeable in FACS analysis and understands that since each WW sample takes many hours to sort and resolves a few hundred thousand cells, the task would be immense, unfeasible. To accommodate with that limitation and to clarify, we toned-down the connection between the FISH-FACS findings and the networks. It is now mentioned that the FISH-FACS data provide strong support to the network analysis as the interaction partners are validated.

The density/frequency dependent fluctuations/oscillations of prey and predator as emphasized in the abstract are not well presented in the manuscript - oscillations are mentioned in connection with the network analysis but hard to connect to density dependent fluctuations - as these co-occurrence patterns could be due to environmental selection or simply driven by spurious correlations. How the physical and chemical

conditions may result in observed co-occurrence patterns needs to be considered and potential biases discussed in much more detail.

We amended the abstract. It now mentions "... can regulate prey populations, possibly in a density-dependent manner" (Line 25), and "as BALO strains differed in potential prey range" (Line 26).

Please pay attention that at no place in the manuscript do we link co-occurrences/ potential predator-prey interactions to environmental variables. We did however relate the environmental parameters to the density of the predators. As mentioned, only temperature was statistically significant, and could explain the dynamics of some of the BALOs. Exactly because no other variable matched BALO dynamics, we hypothesized that prey availability may be a strong driver and may— at least partly — explain the observed fluctuations in relative abundance of BALO OTUs. This is explained in lines 132-141.

We certainly do not claim that the dynamics of ALL gram-negative populations are driven by BALO predation, far from that. On the contrary, a non-biased, all in all computation of the correlations between predator and gram-negative populations using stringent parameters to avoid false positives, sieved out >99.5% of the possible connections between OTUs (Table S5) but showed that the dynamics of specific prey populations are highly correlated with those of specific predator. The point is now made at lines 148-151, along with additional mention of the potential nature of the findings "The analysis, based on significant negative correlations between the three annual datasets (total bacteria, Bdellovibrionales and Bacteriovoracales) (<-0.7, p-value<0.05) sieved >99.5% of possible connections, revealing the co-occurrence of short time scale oscillations between predators and gram negative populations, exhibiting patterns akin to predator-prey cycles, defining potential predator-prey interactions".

Please also note that the 15 environmental variables measured (Table 2) in each sample in each WWTP are those used by WWTP operators and researchers to characterize the quality of the treatment process and thus to describe the environment of a secondary treatment reactor. Surely, they only catch that much of the total variance and the issue of unmeasured and unmeasurable environmental effects is still looming over the field of microbial ecology. Accordingly, we added a sentence mentioning other possible sources of variation "Although unmeasured variables may contribute to about 50% of the variance in WWTP sludge communities, some studies showed the effect of particular environmental variables on BALOs." (Lines 348-349).

As pointed out by one of the reviewers earlier there is no data and analysis presented that would allow to make inferences on density dependent predatory-prey interactions. The results might be hidden somewhere but in the current manuscript this is not conveyed. Maybe focusing on the co-occurrence patterns where you have indications on predatory-prey interactions from the cell sorting results might provide relevant starting points.

In the first paragraph, the reviewer writes " The co-occurrence networks only provide potential predatory-prey interactions". We interpret this as accepting their validity for inferring potential predator-prey interactions. Thus we do not understand what is meant by "no data and analysis presented that would allow to make inferences on density dependent predatory-prey interactions" as the co-occurrence networks are exactly that. Indeed, the co-occurrence patterns are "potential predator-prey patterns". This has now been clarified at all relevant places in the text.

As to "focusing on the co-occurrence patterns where you have indications on predatory-prey interactions from the cell sorting results might provide relevant starting points" *please see our answer on its feasibility above.*

Here some more comments:

L 69 You may use "potential prey"

The sentence has been altered to answer the request and is now "Here, we specifically analyzed the community dynamics of the two major BALO clades (the Bdellovibrionales and the Bacteriovoracales) and their potential association with prey over a year, at three wastewater treatment plants (WWTPs). We further demonstrate the in-situ association of BALOs with identified prey, and invoke predator-prey theory to explain our results."

L 70 What ecological theory are the authors inferring here. They should be more specific.

"Predator-prey theory". Corrected.

L 74 What results allow to make inferences on the co-existence of diverse OTUs of BALOs?

The results supporting niche partitioning are at lines 187-188 and figure 6.

L 76 I am missing results on the microhabitat differentiation. The authors do not provide any results on this - The analyses performed and presented do not allow to make a very strong point as they do not allow to resolve differential distribution across microhabitats.

Thank you for the comment. The sentence has been corrected to " We hypothesized that niche differentiation sustains predator diversity by prey range partitioning, temporal differentiation and through fluctuating predator and prey populations. Finally, in order to validate sequence-based computing results, we developed a direct approach based on FISH tagging and cell sorting to identify the interacting predators and prey. " (Lines 74-79).

L 164-187 The authors have no proof for predation - they have negative correlations that may provide potential prey-predator interactions. This needs to be clarification! The authors can only talk about potential predator-prey interactions.

In accordance with the reviewer's request, we added "potential" where ever relevant to describe these results. Both Predator-prey interactions: networks and Predator-prey interactions: prey range sections have been renamed Co-occurrence networks and Potential BALO prey ranges, respectively.

L 250. How do the authors show dependence upon prey density on shorter time frames? And how is co-existence of numerous predators explained by prey range partitioning? The authors have some indications from the networks based on temporal dynamics but nothing more.

The reviewer is right that these are not classical (Lotka–Volterra type) density-dependent relationships, yet as shown in figure 4, the co-occurrence patterns do suggest density dependence. Surely the systems is far too noisy and certainly includes other factors affecting the dynamics of both predator and prey populations, Accordingly, this has been toned-down with line 253-254 now being " ...upon seasons on the long term and possibly upon prey density on shorter time frames and; appears to explain co-existence of numerous predators by prey range partitioning."

Niche differentiation is proposed to explain the co-existence of so many predators. If they would all compete for the same prey at the same time in the same environment, theory predicts -and experiments show- that this diversity would not be sustainable. However, if the different predators can prey on different prey – even if their prey ranges overlap- their can co-exist. We show that individual BALO OTUs have potential different prey ranges. Yet, as the FISH-FACS experiment validated the identify of potential prey, the niche differentiation hypothesis is accordingly supported. Please see the previous paragraph, showing how it is now toned-down in the opening discussion statement.

L 288. Here only two habitats were differentiated. Could there be others?

Yes, there could be myriads of other microhabitats. As examples, flocs themselves can be differentiated into many different subtypes; there are the surfaces of the reactors, of small eukaryotes (rotifers, daphnias, nematodes, or fungi) were biofilms can form. This has been incorporated in the text, lines 293-295 " However, spatial structure in the form of floc and liquor microhabitat did not segregate BALOs, as the Bd and the Bx communities did not differ between the floc and the liquor microhabitats. This is in contrast to the micro-eukaryote predators and the general bacterial community²³. It may yet be that biofilms on constructed or biological surfaces (e.g. rotifers, daphnias, nematodes, or fungi) provide additional microhabitats for BALOs and prey to interact."

L 305. What were the absolute numbers? Do the authors have some estimates?

Yes, they are shown in figure S2. The BALOs population sizes are about 10^{7-8} in the flocs and 10^{5-6} (per ml) in the liquor.

L 436 Please state that this is the analysis used for bacterial community analysis in the case of 2 WWTPs.

The analysis was performed on three WWTPs, as mentioned in L. 427 and in table S7.

Figure 3. If these are relative abundances based on read numbers this should be stated in the figure description. And units should be given in the figures.

This is now mentioned in the caption. There are no units for relative abundance.

Figure 6. Is there a way how this can be combined with showing the overlaps in potential prey among the different treatments. Please add how these results were obtained.

More details on the combinations of Bd and Bx with the different treatment plants, flocs and liquor are presented in figure S4 and S5. The analysis was performed using genevenn

<http://genevnn.sourceforge.net/index.htm>). This is now mentioned in materials and methods, lines 509-510.

Reviewer #4 (Remarks to the Author):

This manuscript describes an investigation of the population dynamics and interactions of the predatory bacteria, Bdellovibrio and like organisms (BALOs), specifically Bdellovibrionales (Bd) and Bacteriovorales (Bx), in waste water treatment plants. The authors report that the results of their study show greater diversity of the predators than previously known; the population dynamics and interactions of the BALOs with other bacteria in wastewater treatment facilities, the impact of seasonal changes and prey abundance and existence of numerous predators by prey range partitioning. The manuscript has been previously reviewed and the reviewers provided extensive comments. The current review focuses on the authors' responses to the original reviewers' comments and other aspects of the manuscript. The manuscript describes a rather complex study that utilizes many state-of-the-art technologies to answer some critical questions on the interaction of BALOs with other bacteria. The use of flow cytometry to enumerate and determine BALO association with individual cells is a novel approach for the study of the predators. Network analysis was also utilized. These technologies are not without limitations in deciphering BALO interactions, but do provide strong, but presumptive, information based on the experimental data. Nevertheless, the data and the conclusions based on them represent an important advance in the study of BALOs.

The authors' responses to the original reviewers' comments were complete and thorough. The authors did take those comments seriously and in many cases made revisions to the manuscript, and in some cases included results from additional experiments. Although in their responses to the original reviewers' comments, the authors did acknowledge some limitations in some of the methodologies, it would be helpful if they acknowledged more the presumptive nature of their findings and discuss future studies and methodologies to confirm results where appropriate.

We thank the reviewer for her/his appreciation of our work.

We added "This approach may be developed to precisely identify interacting individual predator and prey, for instance by using specific probes for each, based on computed interactions or by isolating single predator-prey pairs (bdelloplast) and performing high throughput single cell sequencing or by directly linking interacting pairs using PCR technologies such as epicPCR" (Lines 417-420).

These changes would be minor involving revisions of a few phrases or words. Several examples are below.

The manuscript has been revised accordingly as this was also a request from another reviewer.

The first is on line 250: Suggest that "uncovers" be changed to suggest or indicate.

Line 250: uncovers differential temporal dependence of BALOs upon seasons on the long term and upon and on line 251 add “appears to” as shown below.

Line 251: prey density on shorter time frames, and; appears to explain co-existence of numerous predators by prey range partitioning.

Corrected (Line 252).

Another example is shown in line 256.

It's suggested that “precise” be deleted.

Finally, the study establishes a

Line 256: novel approach that enables the precise tracking and identification of interacting predators

Corrected

A few other comments follow.

In line 437- Green et al., 2015 is missing from reference list. Please check all the references.

Corrected

Why for some sites they used only BALO specific 16S rRNA gene primer but for the some they used 16S rRNA for total bacteria and BALO both?

Our main aim was sequencing and most of the DNA was used for this purpose. When there was not enough DNA to cover a series we opted for not analyzing it rather than obtaining partial results which are too often difficult to interpret.

The authors should provide the version of the SILVA

Corrected

We hope this answers all queries and comments. Thank you again,

On the behalf of all authors,

Edouard Jurkevitch

REVIEWER COMMENTS

Reviewer #3 (Remarks to the Author):

The authors have done a great job revising the manuscript. Still, I have some minor comments.

L 116. Please report p-values the correct way. <https://scc.ms.unimelb.edu.au/resources-list/understanding-empem-values/report-on-p-values>

I am still missing a comparison between the OTUs/ASVs identified by the sorting and the potential prey OTUs/ASVs correlating in negative fashion with the BALOs. I would like to see some numbers on how many of the sorting identified sequences were also identified by the network analysis. It would be also nice to know if there are some that show positive and none-significant co-occurrence patterns. This is important for validating the network approach.

Here are our point-to-point answers:

L 116. Please report p-values the correct way. <https://scc.ms.unimelb.edu.au/resources-list/understanding-empem-values/report-on-p-values>

The p values are now reported as $p < 0.001$. The size of the difference between sample groups (i.e. WWTP locations) was represented by the A-statistic of the MRPP test (the chance corrected within-group agreement) while its significance was represented by the MRPP P-value. Values above 0.1 are considered high in ecology.

This has been clarified in the text (lines 113-114).

According to McCune and Grace (2002), the following should be reported when using MRPP:

- *A paper providing the details of the methods (Berry and Mielke, 1984, ref.88)*
- *The software used (MjM Software, line 485)*
- *The distance measure (Bray-Curtis, line 486)*
- *How groups are defined and their sizes (e.g. WWTPs, number of samples)*
- *Chance-corrected within group agreement (A-value) and the associated p-value (Figure S3).*

I am still missing a comparison between the OTUs/ASVs identified by the sorting and the potential prey OTUs/ASVs correlating in negative fashion with the BALOs. I would like to see some numbers on how many of the sorting identified sequences were also identified by the network analysis.

Thank you for asking for the clarification. That enabled us to precisely recalculate the overlapping prey OTUs. It is now stated as:

“More specifically, 64 (23.5%) and 46 (10.8%) of the OTUs co-sorting with BALOs, i.e. OTUs of the prey populations and identified in the P1 and P14 samples respectively, overlapped with potential prey OTUs of the LB networks (Table S6).” (lines 245-247).

It would be also nice to know if there are some that show positive and none-significant co-occurrence patterns. This is important for validating the network approach.

The data presented in table S5 include what the reviewer asks for. The total number of connections includes all types of interactions, i.e. not significant ones according to our threshold as well as the significant ones. The % edges/connections are the interactions that are inferred as significant and are thus potential predator-prey interactions. Part of the table is shown below, for convenience.

Total no. of connections	1106672	1133650	1112636	1042944	1257648	1228539
Edges (<=-0.7)	292	113	516	830	399	537
Edges/Connections (%)	0.026	0.010	0.046	0.080	0.032	0.044
Nodes (Predator) (Bd/Bx)	70 (44/26)	49 (32/17)	67 (41/26)	77 (51/26)	65 (36/29)	83 (41/42)

REVIEWERS' COMMENTS

Reviewer #3 (Remarks to the Author):

L230 MDA was mentioned the first time. Please spell out.

L334 Same with KtW models. Spell out the first time.

Concerning the KtW model. Is it applicable if there are multiple preys? Or can the dynamics be less predictable? Please clarify.

Figure 6 could eventually be moved to the supplementary material.

Reviewer #3 (Remarks to the Author):

L230 MDA was mentioned the first time. Please spell out.

Done (L223).

L334 Same with KtW models. Spell out the first time.

Done (Line 325).

Concerning the KtW model. Is it applicable if there are multiple preys? Or can the dynamics be less predictable? Please clarify.

The matter is already covered at lines 343-347 "Yet, KtW has been extended to include more realistic situations than when first proposed, by taking into account the selective grazing of prey by protozoa resulting in different population groups, and interactions of multiple predators with multiple prey⁵³. BALOs similarly have differential access to prey as they have different prey ranges that potentially vary from restricted to generalist^{11,54,55}..."

Figure 6 could eventually be moved to the supplementary material.

We prefer to keep it as a main figure as it illustrates an important point in our results.